# Altering mammalian transcription networking with ADAADi: An inhibitor of ATP-dependent chromatin remodeling

**Radhakrishnan Rakesh[1], Upasana Bedi Chanana[1], Saddam Hussain[1], Soni Sharma[1], Kaveri Goel[1], Deepa Bisht[1], Ketki Patne[1], Pynskhem Bok Swer[1], Joel W. Hockensmith[2]\*, Rohini Muthuswami[1]¤\***

**1** Chromatin Remodeling Laboratory, School of Life Sciences, JNU, New Delhi, India, **2** Department of Biochemistry and Molecular Genetics, University of Virginia, Charlottesville, Virginia, United States of America

¤ Current address: School of Life Sciences, JNU, New Delhi, India
\* rohini_m@mail.jnu.ac.in (RM); jwh6f@virginia.edu (JWH)

**Data Availability Statement:** All RNA seq data (Supplementary Excel File 1) is available from the NCBI GEO database (accession number

## Abstract

Active DNA-dependent ATPase A Domain inhibitor (ADAADi) is the only known inhibitor of ATP-dependent chromatin remodeling proteins that targets the ATPase domain of these proteins. The molecule is synthesized by aminoglycoside phosphotransferase enzyme in the presence of aminoglycosides. ADAADi interacts with ATP-dependent chromatin remodeling proteins through motif Ia present in the conserved helicase domain, and thus, can potentially inhibit all members of this family of proteins. We show that mammalian cells are sensitive to ADAADi but with variable responses in different cell lines. ADAADi can be generated from a wide variety of aminoglycosides; however, cells showed differential response to ADAADi generated from various aminoglycosides. Using HeLa and DU145 cells as model system we have explored the effect of ADAADi on cellular functions. We show that the transcriptional network of a cell type is altered when treated with sub-lethal concentration of ADAADi. Although ADAADi has no known effects on DNA chemical and structural integrity, expression of DNA-damage response genes was altered. The transcripts encoding for the pro-apoptotic proteins were found to be upregulated while the anti-apoptotic genes were found to be downregulated. This was accompanied by increased apoptosis leading us to hypothesize that the ADAADi treatment promotes apoptotic-type of cell death by upregulating the transcription of pro-apoptotic genes. ADAADi also inhibited migration of cells as well as their colony forming ability leading us to conclude that the compound has effective anti-tumor properties.

## Introduction

Epigenetic modulatory proteins include the ATP-dependent chromatin remodeling proteins which have been shown to participate in transcriptional regulation of gene expression by

GSE137251).The ChIP seq data is also available from the GEO database (accession number GSE137250).

**Funding:** R.M. was supported by grants from CSIR (37/(1489)/11/EMR-II, https://www.csirhrdg.res.in/), India as well as from UPE-II, DST-PURSE and DBT-BUILDER. S.S., K.G., S.H., D.B., and K.P. were supported by fellowship from CSIR. U.B.C was supported by HRD fellowship-Young Scientist from the Ministry of Health and Family Welfare. P.B.S. was supported by fellowship from SERB and R.R. was supported by UGC non-net fellowship. The funders had no role in study design, data collection and analysis, decision to publish, or preparation of the manuscript.

**Competing interests:** The authors have declared that no competing interests exist.

harnessing energy released by ATP hydrolysis to reposition nucleosomes [1]. These proteins have been classified into 24 sub-families based on phylogenetic analysis [2].

Experimental evidence has shown that many of these proteins regulate DNA repair as well as gene expression [3]. Mutations and/or dysregulation of expression of these proteins result in manifestation of disease condition. For example, mutations in SMARCAL1 causes Schimke Immunoosseous dysplasia (SIOD) while Coffin-Siris Syndrome is associated with BRG1. BRG1 and BRM, transcriptional regulators, have also been shown to be mutated or mis-regulated in many cancers [4–6]. BRG1 has also been shown to interact with BRCA1, thus, linking it to breast cancer [7]. Loss of expression of BRG1/BRM has been correlated with poor prognosis in lung cancer [8].

While the epigenetic modulators have emerged as potential drug targets and many inhibitors for the histone-modifying enzymes are known, the inhibitors for the ATP-dependent chromatin remodeling proteins are limited in number. Some examples include: PFI-3 targeting the bromodomain of BRG1; orally active small molecule allosteric inhibitors that target the ATPase domain reported by Novartis; and small molecule inhibitors identified by high-throughput screening that target the bromodomain [9–11]. In addition to these inhibitors, we had earlier reported that the enzyme aminoglycoside (neomycin) phosphotransferase catalyzes the formation of a small molecule called Active DNA-dependent ATPase A domain inhibitor (ADAADi) that effectively inhibits the ATPase activity of the ATP-dependent chromatin remodeling proteins [12].

The parent aminoglycosides are known to inhibit protein translation by binding to ribosomes [13] but are not inhibitors of chromatin remodelers [12]. Aminoglycoside phosphotransferases transfer a phosphate group to the 3′-OH of aminoglycosides, thus generating phosphoaminoglycosides resulting in inactivation of aminoglycosides [14]. Our studies have shown that in addition to phosphoaminoglycosides, the aminoglycoside phosphotransferases also catalyze the formation of another small molecule that has been termed as ADAADi [12, 15]. ADAADi can be synthesized from myriad aminoglycoside substrates [15].

ADAADi-induced inhibition is specific to ATP-dependent chromatin remodeling proteins and does not inhibit DNA-independent ATPases [12]. Earlier studies showed that ADAADi can inhibit the ATPase activity of Active DNA-dependent ATPase A Domain (ADAAD), as well as that of human SWI/SNF complex and of Mot1 from *S. cerevisiae* [12]. In addition, these studies established that by inhibiting the ATPase activity, the nucleosome remodeling activity is also affected [12]. Later studies established that ADAADi binds to the ATP-dependent chromatin remodeling proteins through motif Ia of the conserved helicase domain and induces a conformational change that precludes ATP hydrolysis by these proteins in the presence of their DNA effector even though the interaction with ATP and DNA is not impeded [15]. Thus, ADAADi is a small molecule inhibitor generated from aminolygcosides that inhibits the ATPase activity, and consequently the nucleosome remodeling activity of ATP-dependent chromatin remodeling proteins.

Disruption of nucleosome remodeling activity is consistent with the sensitivity of mammalian transformed cell lines like Neuro-2A and MCF-7, which express SMARCAL1 and BRG1, to ADAADi [15, 16]. A recent study has shown that ADAADi targets BRG1 within the cell and therefore, can be used as a potential BRG1 inhibitor [16, 17].

As ADAADi inhibits ATP-dependent chromatin remodeling proteins in vitro and has activity towards cells, we have in this paper sought to further characterize the effect of ADAADi on transformed cells. We demonstrate that a host of mammalian cell types are sensitive to ADAADi with sensitivity to the inhibitor varying by cell type. Further, a given cell line may show differential sensitivity to ADAADi depending on the choice of aminoglycoside used to generate the inhibitor. ADAADi treatment was found to alter the expression of genes

encoding ATP-dependent chromatin remodeling proteins and DNA damage response proteins. In addition, the expression of genes encoding for pro-apoptotic and anti-apoptotic proteins was also found to be altered correlating with increased apoptosis. We also demonstrate that ADAADi inhibits cell migration as well as anchorage-independent growth.

## Material and methods

### Chemicals

All chemicals were of analytical grade and were purchased from Sigma-Aldrich (USA), Qualigens (India), Merck (India), Himedia (India) and SRL (India). LB media was purchased from Himedia (India). $Ni^{+2}$-NTA resin, Protein G beads and Taq polymerase were purchased from Merck-Millipore (USA). cDNA synthesis kit was purchased from Thermo-Fisher (USA). Biorex-70 resin was purchased from Bio-Rad (USA). Cell culture chemicals were purchased from Sigma-Aldrich (USA) while cell culture plasticware was from Corning (USA). SYBR Green mix was purchased from Kapa Biosystems (USA).

### Antibodies

SMARCAL1 antibody was raised against N-terminus HARP domain by Merck (India) (Catalog # 106014). BRG1 (Catalog# B8184) and β-actin (Catalog# A1978) antibodies were purchased from Sigma-Aldrich (USA).

### Primers

The qPCR primers were synthesized by Sigma-Aldrich (USA) and the list of primers used in this study is provided in S1 and S2 Tables.

### Synthesis and purification of ADAADi

ADAADi was synthesized and purified as explained previously in Dutta *et al*. [15].

### Cell lines

All cell lines used in this study were purchased from NCCS, Pune, India and maintained in Dulbecco's Modified Eagle Medium (DMEM) supplemented with 10% fetal bovine serum (FBS), 1% penicillin-streptomycin-amphotericin cocktail and 3.7 g/L sodium bicarbonate (pH 7.4) at 37˚C in the presence of 5% $CO_2$.

### Cell viability assay

5000 cells were seeded in each well of a 96-well plate containing 200 μl media and incubated overnight. The media was replaced with media containing varying concentrations of ADAADi derivatives. The cells were incubated at 37˚ C for the indicated time point. The media in each well was then replaced with media containing 0.45 mg/ml MTT and incubated for 2 hr at 37˚C in a $CO_2$ incubator. The media containing MTT solution was discarded and the purple precipitate formed was dissolved in 100 μl isopropanol. The plate was incubated for 15 min at room temperature and the absorbance was measured at 570 nm.

### Calculation of $IC_{50}$ values

$IC_{50}$ values were calculated using https://www.aatbio.com/tools/ic50-calculator/ ("Quest Graph™ IC50 Calculator." *AAT Bioquest*, *Inc*, 08 Jan. 2021, https://www.aatbio.com/tools/ic50-calculator). This calculator uses a four parameter regression model to calculate $IC_{50}$. The

equation used by the software is as follows:

$$Y = Min + ((Max\text{-}Min)/(1 + (X/IC_{50})^{Hills\ Coefficient})$$

Where X = concentration of inhibitor, Y = % cell viability, Min = minimum of the curve, and Max = maximum of the curve.

## RNA extraction and cDNA preparation

RNA was extracted from untreated and treated HeLa cells as explained in Sharma *et al.* [18]. Briefly, 2–4 μg of isolated RNA was mixed with 0.2 μg oligo dT or random hexamer primer and the volume was adjusted to 12 μl with nuclease-free water. The mix was incubated at 65°C for 5 min and then the temperature was reduced to 40°C. At 40°C, 8 μl of the master mix containing reverse transcription buffer, 1unit of Ribolock, 10 mM dNTP and 0.5 units reverse transcriptase was added and mixed uniformly. The tubes were incubated at 25°C for 5 min (for random primer) and then 42°C for 60 minutes. After incubation, the reaction was incubating at 70°C for 15 minutes to inactivate the reverse transcriptase enzyme.

## qPCR

The reaction was performed in an Applied Biosystems 7500 Real-Time PCR System. Each reaction consisted of 5 μl KAPA SYBR FAST qPCR Master Mix (2X), 0.3 μl cDNA, 10 pmole (1 μl of 1μM) of RT-PCR primers. The reaction volume was adjusted with nuclease free water to 10 μl per reaction-per well. The reaction condition was- (95°C, 20 sec) 1 cycle; (95°C, 3 sec, 60°C, 30 sec (read phase)) 40 cycles. The reads were normalized and quantified with method of ΔΔCt. ΔCt was calculated by (Ct value (gene)—Ct value (housekeeping gene)), then ΔΔCt was calculated by (ΔCt value (treatment)—ΔCt value (control)). Then the fold change was calculated by 2−ΔΔCt.

## PolyA mRNA purification

Poly(A) mRNA was purified using NEBNext® poly(A) mRNA magnetic isolation module (Catalog # E7490S). The purified poly(A) mRNA supernatant was transferred to nuclease free PCR tubes and the eluted poly(A) mRNA was quantified using NanoDrop™ 2000 spectrophotometer.

## RNA-seq library preparation

RNA-seq library was prepared using 100 ng purified poly(A) mRNA using NEBNext Ultra II RNA Library Prep Kit using the protocol provided by the manufacturer. The sequencing was done by Genotypic Technologies, Bengaluru, India.

## RNA-seq and ChIP-seq analysis

Quality of the RNA-seq data was assessed by FASTQC analysis. Reads were quasi-aligned onto the reference human transcriptome using salmon to obtain the transcript counts. Salmon generated transcript quantification were converted to gene level quantification using tximport (version 1.10.1) package and imported to DESeq2 (version 1.22.2) on R (version 3.5.2) for gene level analysis. Normalisation and differential gene expression analysis was done using DESeq2 and Differential transcript usage analysis was done by rnaseqDTU package. Gene annotation was done using chipseeker package on R.

For Chip-Seq, raw reads were trimmed by trimmomatic (version-0.36.5) to remove adaptor sequence and base quality was checked by FASTQC. Trimmed reads were aligned to human

genome (hg38) by BOWTIE2 with default settings. Aligned files were marked for duplicates by PicardMarkduplicates and filtered on bit wise flags by SAM tools on Galaxy platform. Only paired end reads that were mapped in proper pair were selected for peak calling. Biological replicates of SMARCAL1 and BRG1 were merged in a single BAM file before peak calling. Peak calling was performed by MACS2 (Version 2.1.1.20160309.0) with default settings. Both alignment and peak calling was done on the galaxy platform (https://usegalaxy.org). Gene annotation and gene ontology was done using HOMER and clusterProfiler respectively.

## Western blotting

HeLa cells were grown in 100 mm plates till 60–70% confluent and then treated with 5μM ADAADi for 24 and 48 hr. The cell extract was made by adding equal volume of CHAPS buffer and freeze-thawing 3 times. The cell lysate was kept on rocker for 30 minutes at 4˚C and clarified by centrifuging at 14000 rpm at 4˚C for 10 min. The supernatant was transferred into a fresh tube and the protein concentration was determined using Bradford reagent. 100–150 μg protein was loaded into each well of a 10% SDS-PAGE. The protein bands were then transferred to PVDF membrane by electroblotting. After transfer, the blot was rinsed for 5 min each with 1X TBST five times and then incubated in 2% (w/v) skimmed milk in 1X TBST for 1 hr at room temperature. The membrane was again washed with 1X TBST five times for 5 minutes each and then incubated with primary antibodies with the required dilution overnight at 4˚C. Subsequently, the membrane was washed five times for 5 minutes each with 1X TBST and then incubated for 45 min with secondary antibody conjugated with horseradish peroxidase. The membrane was washed 5 times for 5 min each with 1X TBST and finally rinsed once with 1X TBS. The blot was then developed by enhanced chemiluminescence method.

## Comet assay

Single cell gel electrophoresis was performed as described by Nandhakumar *et al*. [19]. Briefly, HeLa cells were grown till they were 60–70% confluent and then given treatment with 5 μM ADAADi for the desired time period. After appropriate treatment, the cells were washed in cold 1X PBS, trypsinized and resuspended in ice-cold PBS. The cell suspension containing $10^4$ cells/slide was embedded in 100 μl of 1% low-melting agarose (Sigma-Aldrich, USA) in 1X PBS and spread onto microscopy slides coated with 1% normal-melting agarose. The cells were lysed in the lysis solution (2.5 M NaCl, 100 mM EDTA, 10 mM Tris base, 8 g/L NaOH to adjust pH 10; 1% Triton X-100, and 10% DMSO) at 4˚C for 2 hr. Following lysis, the slides were placed for 20 min in a tank with cold electrophoresis buffer (300 mM NaOH, 1 mM $Na_2$EDTA, pH 12.5) and electrophoresed for 30 min at 25 V and 300 mA. The slides were then neutralized with 0.4 M Tris-Cl pH 7.5 and the DNA was stained with 0.5mg/ml ethidium bromide. The slides were analyzed using fluorescence microscope (Nikon) at 10X magnification. Casplab software was used for the analysis of tail moment according to the given formula which is a measure of both the smallest detectable size of migrating DNA (represented by comet tail length) and the number of relaxed / broken pieces (represented by DNA intensity in the tail) [20].

$$\text{Tail moment} = (\text{Tail mean-Head mean}) \text{ X Tail\% DNA}/100$$

## FACS analysis

Cell were grown in 35 mm dishes till they were 60–70% confluent and then given treatment with ADAADi for the desired time period. After rinsing with 1 ml of chilled 1X PBS, the cells

were trypsinized and collected in 1 ml of chilled 1X PBS. The cell pellet obtained by centrifuging at 1200 rpm at 4°C for 10 min was resuspended in 70% ethanol gently using vortex to avoid clumping and incubated for 30 min at 4°C. The cells were washed twice with 1 ml of chilled 1X PBS at 1200 rpm at 4°C for 10 min. Subsequently, the cells were treated with 50 μl of 100 μg/ml stock of RNaseI and then incubated with 200 μl of 50 μg/ml propidium iodide for 45 min at room temperature prior to analysis. Analysis was done using BD FACS Calibur 4°C and Modfit software.

## Immunofluorescence

HeLa cells were seeded and grown till they were 60–70% confluent and then given treatment with 5 μM ADAADi for desired time period. The cells were fixed in methanol: acetone (1:1), permeablized with 0.5% Triton X-100, washed and blocked with 1% BSA in 1X PBS overnight at 4°C. The cells were washed with 1X PBS and incubated with γH2AX antibody (1:120 dilution) for 60 min at room temperature. The cells were washed with 1XPBS and then incubated with secondary antibody solution (1:200 dilution) and Hoechst (1:100 dilution) at room temperature for 60 min. The cells were washed with1 ml of 1XPBS five times, covered with a coverslip using 70% glycerol and analyzed using confocal microscope (Nikon TiE) under a 60X oil immersion objective.

## *In vivo* ATPase activity analysis

HeLa cells were grown in 100 mm dishes till they were 60–70% confluent and then treated with 5 μM ADAADi for the desired time period. The cells were trypsinized and washed with 1X PBS twice followed by incubation on rocker at 4°C for 1 hr in 200 μl lysis buffer (50 mM Tris-Cl (pH 7.5), 400 mM NaCl, 1 mM EDTA, 1 mM EGTA, 0.1% NP-40, 1 mM PMSF, and protease inhibitor cocktail) and then sonicated in water bath (4 cycles; 10 s ON and 50 s OFF). The cell lysate was clarified by centrifuging at 13,000 rpm for 10 min at 4°C. The lysate was pre-cleared using 20 μl protein G beads and the pre-cleared supernatant was incubated overnight at 4°C with ~ 2 μg polyclonal antibodies either against SMARCAL1 or BRG1. 50 μl equilibrated Protein G beads were added to immunoprecipitated protein bound to the polyclonal antibodies. The beads were centrifuged at 2500 rpm for 3 min, the supernatant was discarded, and the beads were washed with lysis buffer four times. The ATPase activity of the immuno-precipitated SMARCAL1 and BRG1 was estimated using NADH oxidation assay at 340 nm as detailed in Dutta *et al.* [15].

## Acridine orange/ethidium bromide staining to quantitate apoptotic cells

The number of apoptotic cells after ADAADi treatment were quantitated using Acridine Orange/Ethidium bromide staining as per the protocol described by Ribble *et al.* [21]. Quantitation and analysis were done as described by Anasamy *et al.* [22].

## Soft agar colony formation assay

DMEM (2X) containing FBS, and antibiotics was mixed with equal volume of agar (1% w/v) at 42°C. To prepare an agar bed, a 6-well plate was plated with 1.5 ml of the DMEM-agar mix and allowed to solidify. HeLa and DU145 cells (either untreated or treated with sub-lethal concentration of ADAADi for 24 hr) were trypisinised and the pellet was mixed with 5 ml of 2X DMEM and 5 ml of 0.6% (w/v) agar. This cell mix (1.5 ml) was plated on top of the agar bed. After a short incubation at room temperature that allowed the gel to solidify and form a mesh around the cells, 1.5 ml of 1X DMEM containing FBS, and antibiotics was added. To study the

effect of ADAADi, sub-lethal concentration of ADAADi was added to the media. The plates were incubated in the $CO_2$ incubator at 37°C, and images were taken on the 4th, 8th, 12th and 16th day using Nikon TiS microscope at 50X, 200X and 400X magnification.

The images were calibrated, and pixels were converted into μm by the Nikon software using default settings. A Region of Interest (ROI) was established using auto-detect ROI on colonies. The software calculated the area for each colony separately and gave values in $\mu m^2$. For each day, the area for approximately 100 colonies was calculated and the data was plotted using Box-Whisker plot.

## Wound healing assay

HeLa cells were grown in 35 mm dish to 80% confluency before adding sub-lethal concentration of ADAADi for 24 and 48 hr. A scratch was made with 10 μl pipette tip in the middle of the dish. The media was aspirated off to remove the dead cells and fresh media was added. The media for the control cells did not contain ADAADi while in the ADAADi treated cells, the fresh media contained sub-lethal concentration of ADAADi. The migration of the cells and the recovery of the wound at every 12 hr time point were monitored using Nikon microscope TiS at 50X magnification.

## Annexin V-FITC apoptosis detection

The cells were grown in 60 mm dish to 60–70% confluency before treatment with sub-lethal concentration of ADAADi for desired period of time. The cells stained with Annexin V and PI using Annexin V-FITC apoptosis detection kit as per the manufacturer's instructions (eBiosciences, USA) (Catalog # BMS500FI/100). The samples were analyzed in BD FACS Calibur 4C flow cytometer.

## Zymography assay

To assess the activity of metalloproteinases, zymography assay was performed. Cells (untreated and ADAADi treated) were grown in serum-free DMEM media for 12 hr and 50 μl of the media was loaded onto an 8% SDS-polyacrylamide gel containing 0.2% (w/v) gelatin.

The samples were electrophoresed at constant voltage (125V) till the dye front reached the end of the gel. The gel was renatured by incubating it in buffer containing 50 mM Tris-Cl, pH 7.5, 5 mM $CaCl_2$, 1 μM $ZnCl_2$, 2% (w/v) $NaN_3$ and 2.5% (v/v) Triton-X100 for 60 min at room temperature. The renatured gel was then transferred to incubation buffer (50 mM Tris-Cl, pH 7.5, 5 mM $CaCl_2$, 1 μM $ZnCl_2$, 2% (w/v) $NaN_3$ and 1% (v/v) Triton-X100) for 24 hr. After incubation, gel was washed and stained with Coomassie Brilliant Blue R-250 for 1 hr and distained in destaining solution. The breakdown of gelatin was monitored as the appearance of white band in the background of blue stain.

## Statistical analysis

Statistical analysis was done using Sigma-plot. Student's t test was employed to calculate p values. A p-value less than 0.05 was considered as statistically significant. The p-value was calculated only if the change was at least 25% as compared to the control.

## Original blots and gels

Original blots and gels of the western blots and zymography gels presented in this study are shown in S8 Fig.

## Results

### Cellular responsiveness to ADAADi is complex including time-dependent cellular exposure

ADAADi was discovered as an inhibitor of ADAAD, the bovine homolog of SMARCAL1. Previous studies have shown that ADAADi can be used effectively to kill breast cancer cells as well as prostate cancer cells [16, 17]. To understand whether ADAADi can be used beyond those derived from hormone sensitive cancer cell lines, the effect of this molecule on nine different cancer cell lines was studied. These cell lines were of different tissue origins but possessed epithelial morphology (S3 Table).

We quickly realized that quantitation of the effect of ADAADi on mammalian cells required an investigation of the timeframe required for its action on cellular processes. Mammalian cell lines were treated with neomycin derived ADAADi (ADAADiN) for 24 hr and 48 hr and the viability of the cells was measured at the two points. The MTT assay showed that the treatment yielded greater alteration of cellular metabolism at 48 hr as compared to 24 hr (S1 Fig). Therefore, all subsequent experiments were done for 48 hr of ADAADi treatment unless otherwise stated. The $IC_{50}$ (Table 1) was calculated using the equation provided in the methods section. In each case, the $IC_{50}$ at 48 hr was lesser than at 24 hr, implying greater inhibition at the later time point (S1 Fig). It should be noted that in case of mechanism-based inhibition, the $IC_{50}$ values depend on the time and therefore, the values are different at 24 and 48 hr [23].

### Responsiveness to ADAADi is a function of both the ADAADi derivative and the mammalian cell line

ADAADi can be generated from many different aminoglycosides. Therefore, the next question asked was whether mammalian cells responded equally to ADAADi derived from different parental substrates. To address this question, mammalian cancer cell lines were treated with ADAADi generated from neomycin (ADAADiN) or kanamycin (ADAADiK) for 48 hr and we found that cell lines displayed differential sensitivity to the inhibitor, for example, in case of DU145 cells (S1B Fig; Table 1). Another example is L-132 cell line that is derived from embryonic lung tissue with expression of HeLa cell line markers, but this cell line is 63-fold more

**Table 1. $IC_{50}$ values calculated for different cancer cell lines with ADAADiN (at 24 and 48 hr) and ADAADiK (48 hr).**

| Cells | IC50 (µM) for ADAADiN | | IC50 (µM) for ADAADiK |
|:---:|:---:|:---:|:---:|
| | 24 hr | 48 hr | 48 hr |
| A549 | 8.6 | 2.6 | 11.9 |
| DU145 | 18.4 | 6.9 | 9.8 |
| PC3 | 20.3 | 10.1 | 8.8 |
| HepG2 | 9.2 | 4.8 | 12.6 |
| HeLa | 24.7 | 10.1 | 10.9 |
| L132 | 19.5 | 11.3 | 686[#] |
| MCF-7 | 18.9 | 3.9[#] | 379[#] |
| MDA-MB-231 | Could not be calculated | 37.9 | 11.9 |
| HEK293 | Could not be calculated | 24.6 | 6.2[#] |

Except [#], all values were calculated using: https://www.aatbio.com/tools/ic50-calculator/.

[#] Calculated using quadratic equation.

resistant to ADAADiK as compared to HeLa even though both the cell lines show similar sensitivity to ADAADiN. Further, PC3 cells were 1.2-fold more sensitive to ADAADiK as compared to DU145 cells while the DU145 cells were 1.4-fold more sensitive to ADAADiN as compared to PC3 cells. Interestingly, HEK293 cells were 4-fold more sensitive to ADAADiK as compared to ADAADiN (Table 1), which was the inverse relationship for other cell lines.

In all the subsequent experiments we treated HeLa cells with sub-lethal concentration (5 μM) of ADAADiN (henceforth referred as ADAADi).

## ADAADi treatment alters the expression of ATP-dependent chromatin remodeling proteins

The expression of ATP-dependent chromatin remodeling proteins was investigated after treatment of HeLa cells with the inhibitor either for 24 hr or 48 hr. The qPCR data showed that, in case of HeLa cells, the genes could be classified into two broad groups. Group I contained genes whose expression was upregulated after 48 hr of treatment with ADAADi (Fig 1A). This group included genes encoding for BRG1 and SMARCAL1. Group 2 contained genes whose expression was downregulated after 48 hr of treatment (Fig 1B). This group included genes encoding for BRM, RAD54, and CHD proteins.

For the qPCR experiments, *GAPDH* was used as internal control. HeLa cells were treated with 5 μM ADAADi for the time point indicated. The data is presented as average ± s.d. of

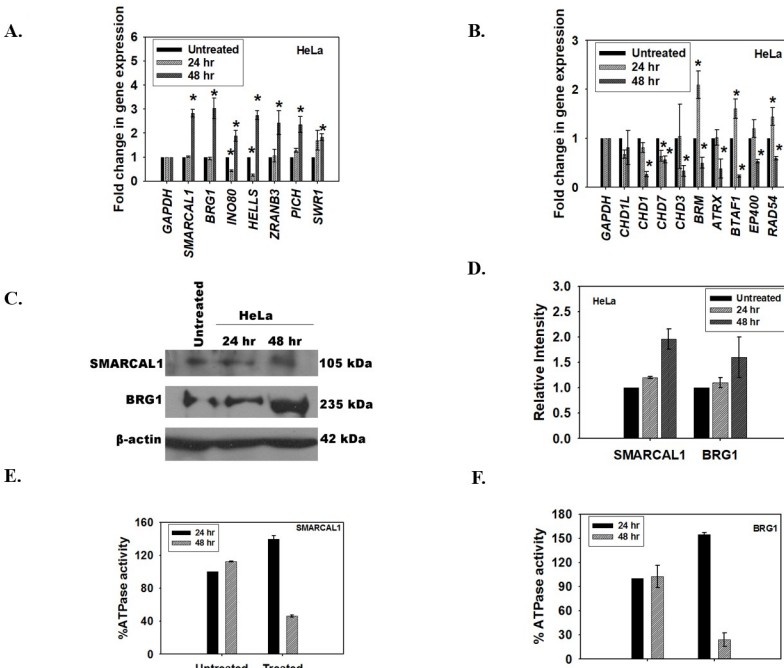

**Fig 1. ADAADi treatment results in rewiring of ATP-dependent chromatin remodeling protein network.** (A). The transcript levels of *BRG1*, *SMARCAL1*, *INO80*, *HELLS*, *ZRANB3*, *PICH*, and *SWR1* was measured using qPCR in HeLa cells after treatment with ADAADi. (B). The transcript levels of *CHD1L*, *CHD1*, *CHD7*, *CHD3*, *BRM*, *ATRX*, *BTAF1*, *EP400*, and *RAD54L* was measured using qPCR in HeLa cells after treatment with ADAADi. (C). The transcript levels of *SMARCAL1* and *BRG1* in HeLa cells after ADAADi treatment were measured using qPCR. (D). Expression of BRG1 and SMARCAL1 in HeLa cells was monitored by western blot after 24 and 48 hr of treatment. (E). Quantitation of the western blot was done using Image J software and normalized with respect to β-actin. (F). The ATPase activity of SMARCAL1 was measured in untreated and ADAADi-treated HeLa cells. (E). The ATPase activity of BRG1 was measured in untreated and ADAADi-treated HeLa cells. In both these experiments, the percent ATPase activity was measure with respect to the untreated control at the respective time points.

three independent experiments. Statistical analysis was done using Sigma-plot and star indicates significant change at p<0.05.

As BRG1 and SMARCAL1 transcripts were upregulated at 48 hr, we confirmed both the protein expression and the ATPase activity of these proteins in HeLa cells after co-immuno-precipitation with respective antibodies. We found that the protein expression was indeed upregulated at 48 hr (Fig 1C and 1D) but the activity was only 46% for SMARCAL1 and 26% for BRG1 at 48 hr indicating that even though the protein expression was upregulated, the resulting protein was inactive (Fig 1E and 1F). As ADAADi is known to bind to ADAAD via motif I and prevent ATP hydrolysis, we hypothesized that even though the protein is produced it is probably inactive due to interaction with the inhibitor molecule [15].

## Global transcriptomic profile of HeLa cells is altered on treatment with ADAADi for 48hr

The alteration of activity of chromatin remodeling proteins led to our hypothesis that the inactivation of the ATP-dependent chromatin remodeling proteins by ADAADi would yield alterations in the transcriptome of cells. To understand the global transcriptomic profile, we treated HeLa cells with ADAADi for 48 hr and performed RNA-seq on the isolated transcripts. As the ATPase activity of the ATP-dependent chromatin remodeling proteins is needed to mediate activities involving chromatin structure, the 48 hr data point was chosen for analysis wherein maximal loss in ATPase activity was observed (see Fig 1E and 1F). The RNA-seq data was analyzed using Salmon, DESeq2 taking UCSC known genes as the reference transcriptome to identify the differentially expressed genes. The data was analyzed using two biological replicates. The data was normalized (S2A Fig) and analysis yielded good correlation as well as mean-variance relationship between the replicates (S2B and S2C Fig). This data is available on GEO database (GSE137251).

About 800 genes that were involved in protein targeting, metabolic processes and protein translation were found to be differentially regulated (p< 0.05) (S1 File). The heat map of the top 100 (p<0.05) differentially expressed genes between the untreated and treated samples showed that there was consistency between the replicates and that there are more downregulated genes as compared to upregulated genes on ADAADi treatment (Fig 2A). A volcano plot was also generated to observe the overall differences between the untreated and ADAADi-treated samples (Fig 2B). Gene ontology showed that the differentially expressed genes regulated protein targeting to ER, ATP metabolic processes, ribosome assembly, and ribosome biogenesis pathways (Fig 2C). Pathway analysis using KEGG also showed that 49 genes involved in apoptosis and 75 genes involved in the cell cycle are differentially expressed upon ADAADi treatment (Fig 2D). *SMARCAL1* and *BRG1* were upregulated, as we had found by qPCR (see Fig 1A), but the change in their expression was not statistically significant (S1 File). Further, the transcripts encoding for ABCG1, ABCB1, and ABCG2 proteins (multi-drug resistance pumps) were found to be unaltered in the case of HeLa cells unlike MDA-MB-231 cells where treatment with ADAADi resulted in reduced expression of these pumps [16].

## ADAADi treatment alters the expression of DNA-damage response genes

As ATP-dependent chromatin remodeling proteins are known to transcriptionally co-regulate the expression of DNA damage response genes, we next queried whether the expression of *ATM*, *ATR*, *Chk1*, *Chk2* and *53BP1* was altered when cells were treated with ADAADi. Of these, ATM and ATR kinases are signal transducers while Chk1, Chk2 and 53BP1 function downstream of these kinases. Further, BRG1 and SMARCAL1 are known to regulate the expression of *ATM* and *ATR* both by ChIP-seq (S4 Table) as well as by experimental data [24].

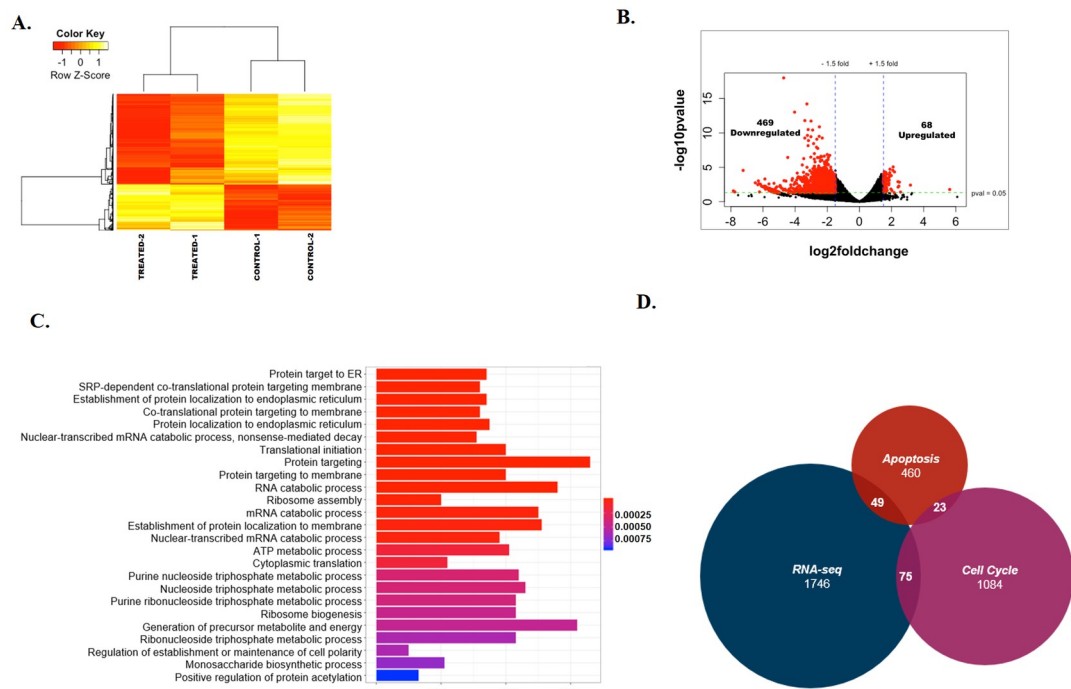

**Fig 2. ADAADi treatment alters transcriptome in HeLa cells.** (A). Heat Map showing the differential expression of genes between untreated and ADAADi treated HeLa cells. (B). Volcano plot showing the differentially expressed genes between untreated and ADAADi treated HeLa cells. (C). Gene ontology showing the various classes of genes that are altered on ADAADi treatment. (D). Intersection between the differentially expressed genes identified by RNA-seq experiment, genes involved in apoptosis and cell cycle.

Quantitation of transcript levels by qPCR showed that the expression of *ATM*, *ATR*, *Chk2* and *53BP1* was significantly altered after ADAADi treatment (Fig 3A). The alteration in the expression of *ATM* transcript was confirmed by RNA-seq also; however, it should be noted that the RNA-seq data showed that this transcript was downregulated while qPCR showed that it was upregulated after 48 hr ADAADi treatment. Quantitation by qPCR showed that the expression of *ATR*, *Chk2* was upregulated, *53BP1* was downregulated and *Chk1* was unchanged upon treatment with ADAADi for 48 hr (Fig 3A).

As the expression of *ATM* was upregulated (by qPCR) after 48 hr of ADAADi treatment in HeLa cells we hypothesized that the cells would show increased DNA damage and therefore, increased γH2AX [25, 26] and 53BP1 foci [27]. Surprisingly, neither γH2AX nor 53BP1 foci were observed in HeLa cells upon ADAADi treatment indicating that DNA damage is not induced in these cells (Fig 3B). This was further confirmed by comet assay (Fig 3C and 3D), leading us to conclude that the alteration in the expression of DNA damage response genes is not correlated with any induction of DNA damage.

## ADAADi treatment results in upregulation of pro-apoptotic and downregulation of anti-apoptotic genes

Cell death can be induced by many mechanisms, including apoptosis. In order to place our transcriptome studies in context, we first confirmed that apoptosis is induced in HeLa using Annexin V staining as well as acridine orange/ethidium bromide staining. ADAADi treatment resulted in an increase in Annexin V positive HeLa cells as compared to the untreated control (Fig 4A and 4B). The acridine orange/ethidium bromide staining method allows for

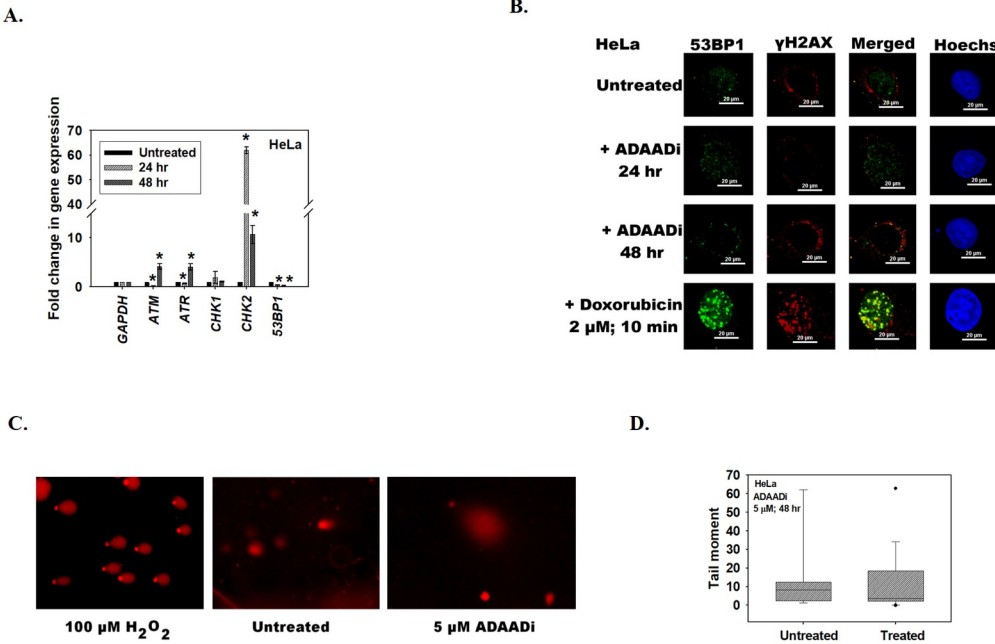

**Fig 3. ADAADi treatment alters the expression of DNA-damage response genes.** (A). The transcript levels of *ATM*, *ATR*, *Chk1*, *Chk2*, and *53BP1* was estimated using qPCR in HeLa cells after treatment with 5 µM ADAADi. In these experiments, *GAPDH* was used as internal control in qPCR experiments. The data is presented as average ± s.d. of three independent experiments. Statistical analysis was done using Sigma-plot and star indicates significant change at $p < 0.05$. (B). Formation of γH2AX and 53BP1 foci was assessed in HeLa cells after treatment with ADAADi (for 24 and 48 hr) using immunofluorescence. For this experiment, 2 µM doxorubicin treatment for 10 min was used as a positive control. (C). Comet assay in the presence of 100 µM $H_2O_2$ treatment for 10 min (positive control), untreated, and ADAADi treated (48 hr) HeLa cells. (D). Quantitation of tail moment in untreated and ADAADi treated HeLa samples.

discrimination between live, apoptotic (early and late) and necrotic cells [22, 28]. The results indicated that the number of apoptotic cells increased after ADAADi treatment as compared to the untreated control (Fig 4C). Concomitantly, there was a decrease in the number of viable cells after ADAADi treatment (Fig 4D). Taken together, these results indicate that the HeLa cells undergo apoptosis upon ADAADi treatment.

Next, we asked whether alteration in the expression of pro- and anti-apoptotic genes on treatment with ADAADi leads to apoptosis. RNA-seq data showed that the expression of 49 genes involved in apoptosis pathway was significantly altered on ADAADi treatment. Quantitation of transcripts using qPCR showed that pro-apoptotic proteins like caspase 3 were upregulated in the presence of ADAADi, while other transcripts encoding caspase 9 and PARP were downregulated at 48 hr. In addition, transcript encoding for the anti-apoptotic protein, Bcl-xl, was found to be downregulated (Fig 4E).

The apoptotic profile is complex and is also known to be regulated by miRNA. Both pro-apoptotic as well as anti-apoptotic miRNA have been identified. For example, miR-133b, miR-16a, and miR-15 have been classified as pro-apoptotic [29, 30] while miR-222, miR-21, and miR-92a have been classified as anti-apoptotic [31–33]. miR-34a regulates apoptosis in a cell-type specific manner and it has been reported as pro-apoptotic miRNA in HeLa cells [34]. miR-17 has been reported as pro-apoptotic in DU145 [35] but as anti-apoptotic in HeLa cells [36]. The expression of miRNA is also known to be regulated by ATP-dependent chromatin remodeling proteins [37]. Therefore, the expression of pro-apoptotic and anti-apoptotic miRNA after ADAADi treatment was analyzed.

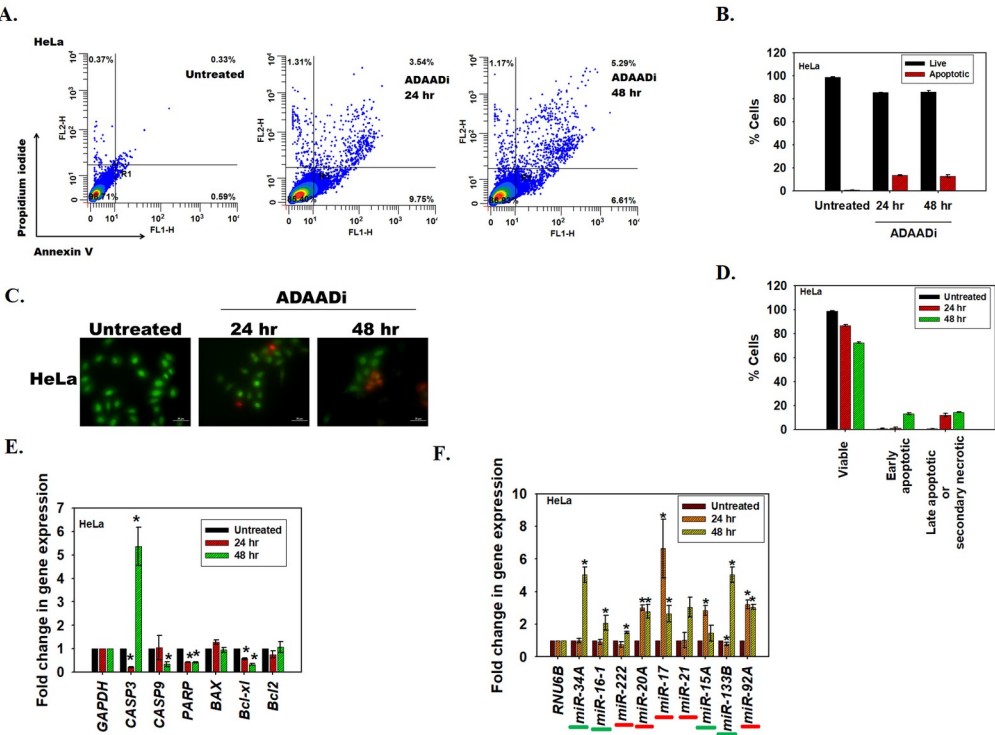

**Fig 4. ADAADi treatment induces apoptosis.** (A). FITC-Annexin V and PI staining in HeLa cells was monitored using FACS in untreated and ADAADi treated cells. (B). Quantitation of the live and apoptotic cells from FACS analysis for HeLa cells. The data is presented as average ± s.d. of three independent experiments. (C). Confocal images obtained after staining untreated and ADAADi treated HeLa cells with acridine orange and ethidium bromide. Green stained cells are viable cells while orange/red cells indicate non-viable or apoptotic cells. (D). The number of viable and apoptotic cells were counted in the untreated and treated HeLa cells. The results are presented as average ± s.d. of two independent experiments. In each experiment, more than 100 cells were counted. (E). The transcript level of *CASP3*, *CASP9*, *PARP*, *BAX*, *Bcl-xl* and *Bcl2* was measured using qPCR in HeLa cells after treatment with 5 µM ADAADi. *GAPDH* was used as the internal control. (F). The levels of pro-apoptotic miRNA (underlined in green) and anti-apoptotic miRNA (underlined in red) were analyzed by qPCR in HeLa cells in the absence and presence of ADAADi. In this experiment, *RNU6B* was used as internal control. The data is presented as average ± s.d. of three independent experiments. Statistical analysis for qPCR experiments was done using Sigma-plot and the star indicates significance at p<0.05.

It was observed that in HeLa cells the pattern of miRNA expression was altered on ADAADi treatment. The pro-apoptotic miRNA, miR-133b and miR-34a, were upregulated after 48 hr of ADAADi treatment in HeLa cells (Fig 4F).

## ADAADi treatment inhibits migration and invasion of cancer cells

With apoptosis established as the mechanism of cell death caused by ADAADi, we turned our attention to another hallmark of cancer-migration and invasion, investigating whether ADAADi could block the invasive property of cancer cells [38]. Therefore, the expression levels of matrix metalloproteinases were investigated as their role in cancer metastasis has been well-established [39]. These proteins are secreted into the extracellular milieu where they promote cancer cell migration by degrading the physical barriers. MMP-2 expression has been shown to be associated with tumor aggressiveness and poor prognosis [39, 40]. ADAADi treatment resulted in decreased expression of *MMP-2* transcripts in HeLa cells (Fig 5A). RNA-seq data also showed that *MMP-2* transcript was downregulated though not in a statistically significant manner. Further, decreased pro-MMP2 secretion in HeLa cells was observed on

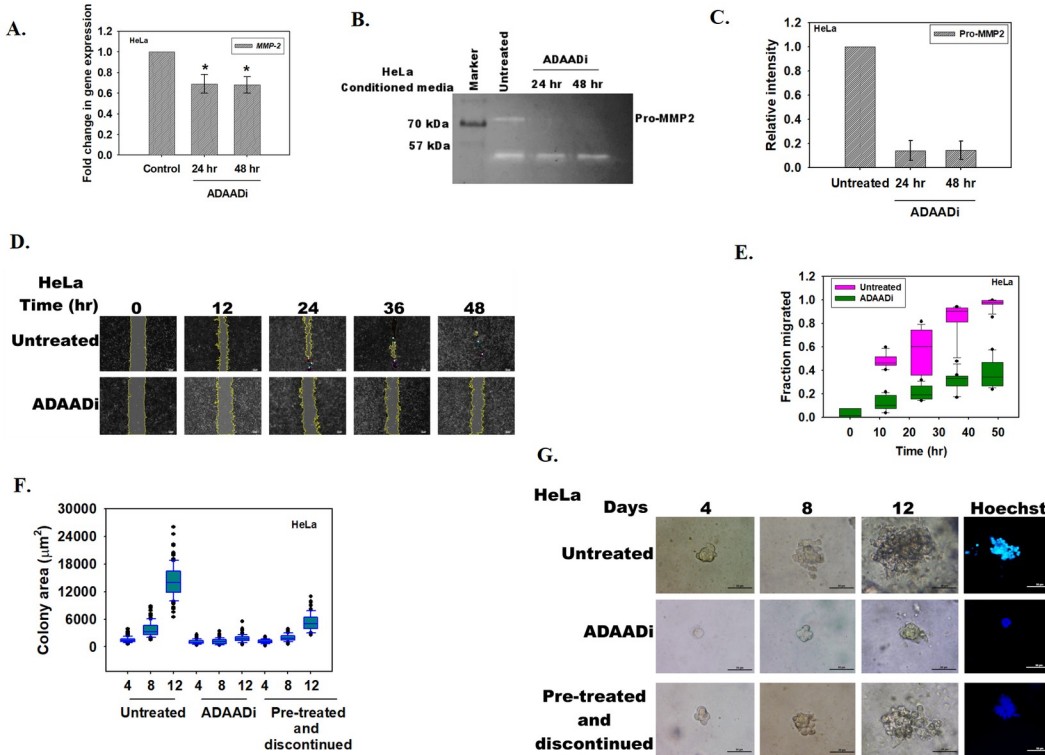

**Fig 5. ADAADi treatment inhibits migration and invasion of cells.** (A). qPCR analysis of *MMP-2* expression in untreated and ADAADi treated HeLa cells. (B). Zymography assay showing the secretion of Pro-MMP2 in the media by HeLa cells. (C). Quantitation of Pro-MMP2 was done using Image J software. The data was normalized with respect to untreated control and is presented as average ± s.d. of two independent experiments. (D). Image analysis of the wound assay captured at different time points after induction of gap in monolayer of HeLa cells. (E). Quantitation of the migration of HeLa cells in the absence and presence of ADAADi as a function of time. The data was normalized with respect to untreated control and is presented as average ± s.d. of three independent experiments. (F). Area (μm²) of the HeLa colonies calculated as a function of time in the absence and presence of ADAADi. (G). Colony formation monitored in untreated, ADAADi treated, and ADAADi pre-treated followed by discontinuation of the inhibitor in HeLa cells. On the 12th day, the colonies were fixed using 100% methanol and stained with Hoechst. Images were taken using Nikon TiS microscope.

ADAADi treatment (Fig 5B and 5C). Concomitantly, wound healing assay showed that the ability to heal the wound was considerably retarded on ADAADi treatment (Fig 5D and 5E).

Taken together, it leads us to conclude that ADAADi treatment has the potential to retard the invasiveness of cancer cells.

## ADAADi inhibits the anchorage-independent growth of cancer cells

Limitless replication is yet another hallmark of cancer cells. We hypothesized that ADAADi should be able to inhibit the ability of cancer cells to replicate indefinitely. To test this hypothesis, soft agar colony forming assay was performed as explained in the Methods section. Two conditions were used to analyze whether ADAADi can inhibit colony formation. In the first condition, ADAADi was added after seeding the cells in the agar containing media. The anchorage-independent growth ability was thus determined in the presence of ADAADi for 12 days. In the second condition, cells were pre-treated with ADAADi for 24 hr. These cells were subsequently seeded in the agar containing media and were allowed to grow in DMEM media that lacked the inhibitor (termed as "pre-treated and discontinued").

ADAADi treatment reduced the soft agar colony forming ability of HeLa as compared to the untreated control when measured by relative area of the colony and Hoechst staining (Fig 5F and 5G). Interestingly, pre-treated and discontinued condition also resulted in decreased colony forming ability, suggesting that pre-treatment of cells with ADAADi is sufficient to block the ability of the cancer cells to replicate indefinitely (Fig 5F and 5G).

## ADAADi treatment induces transcriptomic alterations in DU145 cells

To understand whether the observed phenomena were universal to all cancer cells, the effect of ADAADi on DU145 cells was investigated. These prostate cancer cells, unlike HeLa cells, do not express BRG1 and SMARCAL1 either in the absence or in the presence of ADAADi (S3A Fig). Unlike in HeLa cells, treatment with ADAADi resulted in downregulation of almost all the ATP-dependent chromatin remodeling proteins both at 24 and 48 hr of treatment (S3B and S3C Fig). The exceptions were HELLS which was upregulated and SWR1 which was unchanged both at 24 and 48 hr of treatment (S3B and S3C Fig).

A review of the expression of DNA damage response genes led to the finding that similar to HeLa cells, *ATM* expression was upregulated and *53BP1* was downregulated (Compare Fig 3A and S3D Fig). However, *ATR*, *Chk1* and *Chk2* were downregulated after ADAADi treatment (Compare Fig 3A and S3D Fig). Interestingly, unlike in HeLa cells, DNA damage was induced in DU145 cells as observed by the formation of γH2AX foci and comet assay (S3E–S3G Fig). However, the formation of 53BP1 foci was abrogated indicating DNA damage repair is impaired (S3E Fig). Thus, the ability of ADAADi to induce DNA damage appears to vary from cell line to cell line, which is consistent with our observations that ADAADi inhibition is not limited to a single ATP-dependent remodeler but potentially impacts a matrix of remodeling proteins and their influence of transcription.

As in case of HeLa cells, apoptotic cell death was induced in DU145 cells on ADAADi treatment observed by an increase in Annexin V positive DU145 cells when compared to the untreated control (S4A and S4B Fig). Acridine orange/ethidium bromide staining also indicated that the number of apoptotic cells increased after ADAADi treatment as compared to the untreated control (S4C Fig). Concomitantly, there was a decrease in the number of viable cells after ADAADi treatment (S4D Fig). Taken together, as in case of HeLa cells, DU145 cells also undergo apoptosis.

When the expression pattern of pro- and anti-apoptotic genes were compared between HeLa and DU145 cells, it was observed that the *Bcl-xl* expression was downregulated in both cases (Compare Fig 4A with S4E Fig). Further, in DU145 cells, unlike in HeLa cells, the transcripts of pro-apoptotic genes were also downregulated (Compare Fig 4C with S4F Fig). Similar to HeLa cells, the pro-apoptotic miRNA like miR-133b and miR-34a were upregulated after 48 hr of ADAADi treatment in DU145 cells too (Compare Fig 4C with S4F Fig). The anti-apoptotic miRNA like miR-92a and miR-222 were observed to be downregulated in DU145 cells after ADAADi treatment (S4F Fig).

ADAADi treatment resulted in decreased expression of MMP-2 and MMP-9 transcripts in DU145 cells (S5A Fig). Further, only pro-MMP9 secretion in DU145 cells was observed on ADAADi treatment (S5B and S5C Fig). Concomitantly, wound healing assay showed that the ability to heal the wound was considerably retarded on ADAADi treatment in DU145 cells (S5D and S5E Fig). Finally, ADAADi treatment reduced the soft agar colony forming ability of DU145 cells as compared to the untreated control when measured by relative area of the colony (S5F and S5G Fig).

### The responsiveness of cells is not a function of the status of BRG1 and/or SMARCAL1

Biochemical, biophysical and functional studies have shown that ADAADi targets BRG1 and SMARCAL1 [12, 15]. However, both HeLa (which expresses SMARCAL1 and BRG1) and DU145 cells (express neither SMARCAL1 nor BRG1) showed similar phenotypic effects on ADAADi treatment. Therefore, we asked whether the status of SMARCAL1 and/or BRG1 determine the cell's response to ADAADi? To answer this question, the expression of the two proteins in nine cancer cell lines was analyzed using western blots (S6 Fig). The western blots demonstrated that there was no obvious correlation between the expression of SMARCAL1 or BRG1 and the responsiveness of the cells to ADAADi (Compare S6 Fig with Table 1). For example, BRG1 and SMARCAL1 are both absent in DU145 cells [5] while BRG1 is not expressed in A549 cells [41], while HepG2 and HeLa both express BRG1. Thus, if BRG1 and/ or SMARCAL1 were the sole determinants of a cell's responsiveness to ADAADi, DU145 and A549 should be more resistant to ADAADi as compared to HepG2 and HeLa. However, Table 1 shows that DU145 and A549 are both more sensitive to ADAADi as compared to HepG2 and HeLa, thus suggesting that the presence of BRG1 and/or SMARCAL1 does not determine the responsiveness of a cell line to the inhibitor. As all these cells express other ATP-dependent chromatin remodelers [42, 43] and therefore, the sensitivity of a cell line to ADAADi is most probably determined by the entire repertoire of ATP-dependent chromatin remodelers expressed in that cell.

### ChIP-seq data shows that the transcriptomic changes are not due to BRG1 and/or SMARCAL1

We tested the supposition that transcriptomic changes observed are not solely due to alterations in BRG1 and/or SMARCAL1. We analyzed the available ChIP-seq data (GSE137250), to identify the genome wide occupancy of SMARCAL1 and BRG1 in untreated HeLa cells. Unsupervised k-mean clustering (k = 5) of ChIP-seq peaks showed that the two proteins are present on the transcription start sites (TSS) and their peak overlap showed that these proteins co-occupy 8755 sites on the genome (S7A and S7B Fig).

Since ADAADi acts as a generic inhibitor to ATP-dependent chromatin remodelers, we sought to identify genes where BRG1 and/or SMARCAL1 are present and whether these genes are differentially expressed after the treatment of ADAADi. In this way, we could identify gene targets of ADAADi whose function could be mediated through these two remodelers. To check this, we overlapped genes occupied by SMARCAL1 and/or BRG1 with genes whose expression is altered after ADAADi treatment (1692 genes, padj < 0.09) (S7C Fig). Even though we found that 26.8% (455 genes) of differentially expressed genes are also occupied by either SMARCAL1 and/or BRG1, the overlap was not significant suggesting that these changes are not specifically due to SMARCAL1 and/or BRG1 occupancy (Fisher's exact test, p = 1).

Together, these studies show that ADAADi does not exert its effects solely through SMARCAL1 and BRG1; even though it affects their expression as well as the expression of other chromatin remodelers. Thus, we conclude that the impact of ADAADi is not limited to disruption of function of any single chromatin remodeler but is likely the cumulative sum of the effect mediated by a subset, if not all, of ATP-dependent chromatin remodeler proteins.

## Discussion

The ATP-dependent chromatin remodeling proteins function as co-regulators of transcription in eukaryotic cells. By coupling ATP hydrolysis to nucleosome repositioning, eviction or

histone variant exchange, these proteins are able to regulate the expression of genes, and therefore, the cellular function [1]. Thus, a change in transcriptome by changing the expression of ATP-dependent chromatin remodeling protein can alter the cell fate.

Mammalian cells are sensitive to ADAADi, the inhibitor of ATP-dependent chromatin remodeling protein. Our results indicate that the response a cell line mounts to ADAADi exposure is probably a complex network of expression of the target proteins, the structure of the molecule, the binding affinity of the inhibitor to the target protein, as well as the uptake mechanism. Deciphering the contribution of each of these factors will be required to further understand the differences exhibited by different cancer cell lines towards ADAADi.

ADAADi binds to motif Ia present in ATP-dependent chromatin remodeling proteins [15] and therefore, potentially can inactivate all the proteins belonging to this family. Thus, all cells are likely to be targeted by ADAADi and indeed we have found that all epithelial cancer cells tested could be killed depending on the species of inhibitor and the concentrations employed.

As many of the ATP-dependent chromatin remodeling proteins are transcriptional co-regulators, the transcriptome of the cell is expected to be altered upon treatment with ADAADi. We hypothesize the transcriptome change will depend on the repertoire of ATP-dependent chromatin remodeling proteins being expressed in a particular cell line irrespective of the status of SMARCAL1 and BRG1. Indeed, in our studies we find that the transcriptome of not only HeLa cells (express both BRG1 and SMARCAL1) but also of DU145 cells (does not express BRG1 and SMARCAL1) is altered on ADAADi treatment. The variability in the transcriptome upon ADAADi treatment could be attributable to the subset of ATP-dependent chromatin remodeling proteins being expressed by a given cell line.

By inhibiting the ATPase activity of ATP-dependent chromatin remodeling proteins, ADAADi changes the transcriptome and thus, alters the fate of the cancer cell. The change in the transcriptome of the cell in response to a chemotherapeutic agent is not surprising as it has been reported for many agents including doxorubicin [44, 45] and cisplatin [46]. However, the use of ADAADi brings new perspective to the discussion as the alteration in the transcriptome affects a wide range of properties of cancer cells. Thus, the alteration in transcriptome on ADAADi treatment induces apoptosis, inhibits migratory potential of cells, and inhibits anchorage independent growth of cancer cells. Our work demonstrates that ADAADi treatment results in alterations in the transcriptional network of the cell. The alterations could be classified into i) alterations in the expression of genes encoding for chromatin remodeling proteins; ii) alterations in the DNA damage response genes; iii) alterations in pro-apoptotic and anti-apoptotic genes; iv) alterations in miRNA expression; and v) alterations in the expression of genes involved in migration of cells.

The expression of DNA-damage response proteins was altered on ADAADi treatment; however, in HeLa cell line, DNA damage *per se* could not be detected either by γH2AX staining or by comet assay. The comet assay showed a peculiar feature on ADAADi treatment. The head of the comet was observed to be separated from the tail, a so-called hedgehog comet, an indication that the cells might be undergoing apoptosis [47]. This was further confirmed by Annexin V staining as well as caspase activity indicating that ADAADi is able to induce DNA-damage independent apoptosis in HeLa cells. DNA-damage independent induction of apoptosis has been reported for curcumin [48] though the predominantly observed pathway for induction of apoptosis is through DNA damage. We hypothesize that in case of HeLa cells, the alteration in the transcriptome induces apoptosis. ATP-dependent chromatin remodeling proteins are known to regulate the transcription of many genes including those involved in apoptosis. For example, loss of ATRX is correlated with increased apoptosis during development [49]. Further, the expression of miRNA is also altered on ADAADi treatment. Of importance is the upregulation of miR-34a. Ectopic expression of miR-34a has been shown to promote

apoptosis in neuroblastoma cell lines and indeed, miR-34a has been established as a tumor suppressor [50]. Thus, the upregulation of miR-34a on ADAADi treatment might also contribute to pushing the cell towards the apoptotic pathway.

In contrast, in DU145 cells, DNA damage was observed but DNA repair was impaired as 53BP1 foci could not be observed. We hypothesize that DNA damage response and repair in response to ADAADi differs based on the available set ATP-dependent chromatin remodeling proteins in the cell.

Finally, a major property of cancer cells is their ability to migrate and thus, metastasize to new sites. By rewriting the transcriptome, ADAADi treatment is able to block migration and invasiveness of cancer cells. Thus, the sum of our results provides a compelling case for the chemotherapeutic potential of this novel class of drugs targeting ATP-dependent chromatin remodeling and those DNA metabolic processes essential to cellular integrity.

## Supporting information

**S1 Fig. ADAADi induces cell death in cancer cell lines.** Kill curves showing the effect of ADAADi after treatment. (A). HeLa (B). DU145 (C). MCF-7 (D). L132 (E). PC3 (F). MDA-MB-231 (G). A549 (H) HepG2 (I) HEK293.
(PDF)

**S2 Fig. Analysis of the RNA-seq data.** (A). The counts per sample before and after normalization. (B). Mean-variance relationship between the untreated and ADAADi-treated RNA-seq samples. (C). Correlation between untreated and ADAADi-treated RNA-seq samples.
(PDF)

**S3 Fig. ADAADi treatment induces transcriptomic alterations in DU145 cells.** (A). Expression of SMARCAL1 and BRG1 in untreated and 2 μM ADAADi treated cells at 24 and 48 hr. β-actin was used as loading control. (B). The transcript levels of *INO80*, *HELLS*, *ZRANB3*, *PICH*, and *SWR1* was measured using qPCR in DU145 cells after treatment with ADAADi. (C). The transcript levels of *CHD1L*, *CHD1*, *CHD7*, *CHD3*, *BRM*, *ATRX*, *BTAF1*, *EP400*, and *RAD54L* was measured using qPCR in DU145 cells after treatment with ADAADi. (D). The transcript levels of *ATM*, *ATR*, *Chk1*, *Chk2*, and *53BP1* was estimated using qPCR in DU145 cells after treatment with 2 μM ADAADi. (E). Formation of γH2AX and 53BP1 foci was assessed in DU145 cells after treatment with ADAADi (for 24 and 48 hr) using immunofluorescence. For this experiment, 2 μM doxorubicin treatment for 10 min was used as a positive control. (F). Comet assay in the presence of 100 μM $H_2O_2$ treatment for 10 minutes (positive control), untreated, and ADAADi treated (24 and 48 hours) DU145 cells. (G). Quantitation of tail moment in untreated and ADAADi treated DU145 samples. For the qPCR experiments, *GAPDH* was used as internal control. HeLa cells were treated with 5 μM ADAADi for the time point indicated. The data is presented as average ± s.d of three independent experiments. Statistical analysis was done using Sigma-plot and star indicates significant change at p<0.05.
(PDF)

**S4 Fig. ADAADi treatment causes apoptosis in DU145 cells.** (A). A). FITC-Annexin V and PI staining in DU145 cells was monitored using FACS in untreated and ADAADi treated cells. (B). Quantitation of the live and apoptotic cells from FACS analysis for DU145 cells. The data is presented as average ± s.d. of three independent experiments. (C). Confocal images obtained after staining untreated and ADAADi treated DU145 cells with acridine orange and ethidium bromide. Green stained cells are viable cells while orange/red cells indicate non-viable or apoptotic cells. (D). The number of viable and apoptotic cells were counted in the untreated and treated DU145 cells. The results are presented as average ± s.d. of two independent

experiments. In each experiment, more than 100 cells were counted. (E). (A). The transcript level of *CASP3*, *CASP9*, *PARP*, *BAX*, *Bcl-xl* and *Bcl2* was measured using qPCR in DU145 cells after treatment with 2 μM ADAADi. *GAPDH* was used as the internal control in these experiments. (F). The levels of pro-apoptotic miRNA (underlined in green) and anti-apoptotic miRNA (underlined in red) were analyzed by qPCR in DU145 cells in absence and presence of 2 μM ADAADi. In these experiments, *RNU6B* was used as internal control. Statistical analysis for qPCR experiments was done using Sigma-plot and the star indicates significance at $p < 0.05$.
(PDF)

**S5 Fig. ADAADi treatment inhibits invasion and colony formation in DU145 cells.** (A). qPCR analysis of *MMP-2* and *MMP-9* expression in untreated and ADAADi treated DU145 cells. (B). Zymography assay showing the secretion of Pro-MMP2 in the media by DU145 cells. (C). Quantitation of Pro-MMP2 was done using Image J software. The data was normalized with respect to untreated control and is presented as average ± s.d. of two independent experiments. (D). Image analysis of the wound assay captured at different time points after induction of gap in monolayer of DU145 cells. (E). Quantitation of the migration of DU145 cells in the absence and presence of ADAADi as a function of time. The data was normalized with respect to untreated control and is presented as average ± s.d. of three independent experiments. (F). Area ($\mu m^2$) of the DU145 colonies calculated as a function of time in the absence and presence of ADAADi. (G). Colony formation monitored in untreated, ADAADi treated, and ADAADi pre-treated followed by discontinuation of the inhibitor in DU145 cells. On the 12th day, the colonies were fixed using 100% methanol and stained with Hoechst. Images were taken using Nikon TiS microscope.
(PDF)

**S6 Fig. Expression of SMARCAL1 and BRG1 is altered in cancer cell lines.** Expression of SMARCAL1 and BRG1 was analyzed by western blot in HEK293, L132, HeLa, DU145, PC3, A549, HepG2, MCF-7, and MDA-MB-231. β-actin was used as loading control.
(PDF)

**S7 Fig. The genome-wide occupancy of SMARCAL1 and BRG1 was studied using ChIP-seq.** (A) Heat map of chip peaks (BRG1-blue, SMARCAL1-green) after Unsupervised k-mean (k = 5) clustering. SMARCAL1 and BRG1 shows peak occupancy over TSS on cluster 2. There was no binding over TSS on cluster 3 and 4. (B) ChIP-seq peak intersection between SMARCAL1 and BRG1. (C) Intersection among genes occupied by SMARCAL1, BRG1 and differentially expressed genes (padj<0.1) after ADAADi treatment (Fisher's exact test, p = 1).
(PDF)

**S8 Fig. Original blots and gels.**
(PDF)

**S1 Table. List of primers used for qPCR for mRNA.**
(XLSX)

**S2 Table. List of stem-loop primers used for first strand cDNA synthesis and forward primers used in qPCR for mature miRNAs.** The sequence of universal reverse primer used in the qPCR reactions is `5' GTGCAGGGTCCGAGGT3'`.
(XLSX)

**S3 Table. List of cell lines used in this study.**
(XLSX)

**S4 Table. List of genes where presence of BRG1 and SMARCAL1 was detected by ChIP-seq analysis.** NS indicates Not Significant at padj < 0.09, NA indicates transcript was not identified in the RNA-seq data, ✓ indicates presence, ✗ indicates absence.
(XLSX)

**S1 File.**
(XLSX)

## Acknowledgments

The authors would like to thank the Central Instrumentation facility, School of Life Sciences for confocal microscope and FACS facility. The authors would also like to thank Dr. Sarika Gupta for technical support. We would also like to thank Prof. Sneha Sudha Komath, School of Life Sciences, for her valuable inputs.

## Author Contributions

**Conceptualization:** Joel W. Hockensmith, Rohini Muthuswami.

**Formal analysis:** Radhakrishnan Rakesh, Upasana Bedi Chanana, Saddam Hussain, Rohini Muthuswami.

**Funding acquisition:** Upasana Bedi Chanana, Joel W. Hockensmith, Rohini Muthuswami.

**Investigation:** Radhakrishnan Rakesh, Upasana Bedi Chanana, Saddam Hussain, Soni Sharma, Kaveri Goel, Deepa Bisht, Ketki Patne, Pynskhem Bok Swer.

**Methodology:** Radhakrishnan Rakesh, Upasana Bedi Chanana.

**Supervision:** Rohini Muthuswami.

**Writing – original draft:** Radhakrishnan Rakesh, Upasana Bedi Chanana, Joel W. Hockensmith, Rohini Muthuswami.

**Writing – review & editing:** Saddam Hussain, Kaveri Goel, Joel W. Hockensmith, Rohini Muthuswami.

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
