## [Decision Letter · Decision Letter 0]

7 Jan 2021

PONE-D-20-37077

ALTERING MAMMALIAN TRANSCRIPTION NETWORKING WITH ADAADi: AN INHIBITOR OF ATP-DEPENDENT CHROMATIN REMODELING

PLOS ONE

Dear Dr. Muthuswami,

Thank you for submitting your manuscript to PLOS ONE. After careful consideration, we feel that it has merit but does not fully meet PLOS ONE’s publication criteria as it currently stands. Therefore, we invite you to submit a revised version of the manuscript that addresses the points raised during the review process.

We look forward to receiving your revised manuscript.

Kind regards,

Srinivas Saladi, Ph.D.

Academic Editor

PLOS ONE

Journal Requirements:

'FUNDING

R.M. was supported by grants from CSIR (37/(1489)/11/EMR-II), India as well as from UPE-II, DST-PURSE and DBT-BUILDER. S.S., K.G., S.H., D.B., and K.P. were supported by fellowship from CSIR. U.B.C was supported by HRD fellowship-Young Scientist from the Ministry of Health and Family Welfare. P.B.S. was supported by fellowship from SERB and R.R. was supported by UGC non-net fellowship.'

'R.M.

37/(1489)/11/EMR-II

Council of Scientific and Industrial Research, India

https://www.csirhrdg.res.in/

The funders had no role in study design, data collection and analysis, decision to publish, or preparation of the manuscript.'

Additional Editor Comments:

Thank you for submitting the manuscript on testing ADAADinhibitors to target SWI/SNF complex remodeling activity. Two of the reviewers had major concerns while there is enthusiasm in the overall concept of targeting the chromatin remodeler. We will be happy to consider your manuscript if you are able to address the major points from the reviewers.

Reviewers' comments:

Reviewer's Responses to Questions

**Comments to the Author**

1. Is the manuscript technically sound, and do the data support the conclusions?

Reviewer #1: Partly

Reviewer #2: Yes

Reviewer #3: Partly

2. Has the statistical analysis been performed appropriately and rigorously? 

Reviewer #1: Yes

Reviewer #2: Yes

Reviewer #3: No

3. Have the authors made all data underlying the findings in their manuscript fully available?

Reviewer #1: Yes

Reviewer #2: Yes

Reviewer #3: Yes

4. Is the manuscript presented in an intelligible fashion and written in standard English?

Reviewer #1: No

Reviewer #2: Yes

Reviewer #3: No

5. Review Comments to the Author

Reviewer #1: In this paper, the authors explore the cellular effects of ADAADi, a chromatin remodeler ATPase inhibitor that is a metabolite of aminoglycosides.

I have several suggestions for improvement, but the overarching concern with the manuscript is a general lack of confidence in whether any of the phenotypes/gene changes observed are related to the inhibition of SMARCAL1 and BRG1. The evidence that these compounds inhibit ATPase function of SMARCAL1 and BRG1 is strong, but the fact that the compounds inhibit the growth of cell lines that don’t express SMARCAL1 and BRG1 calls the relevance of this function into question. If the conclusion of the manuscript is that ADAADi is a useful chemotherapy with unknown targets, that would be fine, as long as the data supports that conclusion. The pieces of data that do support that conclusion is the fact that ADAADi treatment doesn’t downregulate SMARCAL1 or BRG1 expression, and the general lack of correlation between gene expression changes (and miRNA changes) with ADAADi and shBRG1 or shSMARCAL1. The only data supporting ADAADi targeting BRG1/SMARCAL1 is the viability data in combination with knockdown; however, even that is problematic as any two agents that decrease cell growth would give similar results. Nevertheless the discussion section still centers around ADAAdi acting as a BRG1/SMARCAL1 inhibitor, which is not supported in any way from the results. If the manuscript aims to define ADAADi as a potentially useful chemotherapy with unknown mechanism, there should be more results and discussion around this point.

Some possible avenues to pursue to create a more cohesive story:

Establish more definitively that ADAADi is not acting as a remodeling inhibitor. This could be through functional analysis (ATAC-Seq, ChIP-Seq, RNA-Seq). For example, I see the authors have performed ChIP-Seq for these factors (GSE137250), as have other labs. How does the RNA-Seq using ADAADi compare with BRG1/SMARCAL1 binding sites? Does ADAADi affect BRG1 binding and chromatin accessibility at these sites? Another possibility is to compare ADAADi with the Novartis BRG/BRM inhibitor (or the BI degrader), which has been shown functionally to affect BRG1 activity genome-wide. I know they are providing the compound for collaboration with academic labs (although with lots of MTA and hoops to jump through)

If the conclusion is that it is most likely not acting as a remodeling inhibitor, use the RNA-Seq data to develop hypotheses around what it is doing in cancer cells. There are many databases available for correlation to other inhibitors/processes. Are they affecting translation? The RNA-Seq indicates they might be. What is the activity of these metabolites on the ribosome?

If the conclusion in the end is that they are good chemotherapies, I would add additional non-transformed cell lines to support this possibility.

Minor concerns

There are many more inhibitors of the SWI/SNF chromatin remodeling complexes now. While many of them inhibit chromatin targeting, there are two that act directly on BRG1: the ATPase inhibitor from Novartis, and the BRG/BRM degrader from BI. These two, at the very least, should be cited.

A panel displaying the compound structures referred to in the text (ADAADiN and ADAADiK) would be helpful.

p. 13: It is a little confusing to say the IC50 value at 48 hours is “greater”, when it is actually smaller. Maybe say the inhibition was greater?

Fig 1 D and E, the X-axis should be in log scale.

Reviewer #2: Radhakkrishnan et al report on the genome-wide effects of Active DNA-dependent ATPase inhibitor (ADAADi) in cultured mammalian cells. This inhibitor, derived from neomycin and a related inhibitor, ADAADiK, derived from kanamycin bind to ATP-dependent chromatin remodeling proteins, thereby inducing a conformational change and inhibiting activity. ADAADi was found to inhibit growth of mammalian cell lines, including cancer cells and promote apoptosis. This effect was dependent on BRG1 and SMARCL1 because depletion of these proteins desensitized cells to the drug. Although DNA damage was not observed, the expression of DNA damage response proteins was altered. Gene expression profiling revealed effects on expression of apoptotic protein coding genes and microRNAs. ADAADi also inhibited MMP2 and migration of cancer cells as did depletion of BRG1 and SMARCL1.

Critique: This is the first comprehensive investigation of an inhibitor to the activity of ATP dependent chromatin remodeling enzymes. Although there are concerns about lack of selectivity, the findings that ADAADi can suppress growth and impede cancer migration suggest it may have therapeutic value. The study is well conducted and the findings convincing.

Minor Concerns

1. Figure legend 2B states that Westerns were done on both HeLa and DU145 cells, however, only HeLa cells are shown. Also, the text only indicates HeLa cells.

2. Much of the study associates ADAADi with inhibition of BRG1 and SMARCAL1 activity, yet the abstract does not even mention BRG1 and SMARCAL1. Some mention of the distinct and overlapping effects of ADAADi and depletion of BRG1 and SMARCAL1 by shRNA should be mentioned in the abstract. A little more discussion of this in the Discussion section would also be appropriate.

Reviewer #3: In the present study authors present an overview of the effect of ADAADi on the transcription profile of HeLa cells. Based on the RNA seq analysis of ADAADi treated cells, authors identify upregulation of pro-apoptotic genes as a major feature. They conducted a battery of standard experiments to suggest that ADAADi treatment leads to apoptosis, decrease in cell migration and attenuated cell growth. Based on the observations from their study authors propose that ADAADi has chemotherapeutic potential.

Overall, the study would appeal to the broad audience in cancer research and in principle can be good contribution to the field. However, the manuscript has substantial major issues:

1. Manuscript lacks clarity and does not convey a cohesive story. It lacks features of good organization and there is absence of harmony between the details described in the methods section and the results presented in the manuscript. Authors should ensure that the manuscript is well written with logical flow.

2. Is there any evidence that the use of ADAADi on the non-cancerous cells (eg. fibroblasts) would not illicit similar change in transcriptional profile as being reported here for HeLa cells? If not, then authors should test the effect of ADAADi on transcriptome of non-cancerous cells as well as perform side by side comparison of cancerous vs normal cells in various assays reported here.

3. Most of the experiments have been done using HeLa cells and with only ADAADiN at fixed sub-lethal concentration. However, manuscripts has statements suggesting that ADAADi induced transcriptional rewiring would be a common feature of various cancer cell lines. There is no experimental evidence to propose conservation of such phenomena. It would be interesting to see the effect of ADAADi on transcriptional modulation, and cell growth and migration in a few additional cancer cell lines.

4. In many assays authors are comparing changes in transcript levels for different genes in ADAADi treated cells with the transcript levels of the same genes in cells treated with shRNA against BRG1 & SMARCAL1. Authors conclude that the trends in the two sets were not similar. Is this surprising given that treatment with ADAADi lead to upregulation of both BRG1 and SMARCAL1 whereas shRNA treatment would lead to decrease in level of both the proteins?

5. Does BRG1 and SMARCAL1 regulate the transcription of impacted genes directly or indirectly? Do these chromatin remodelers localize to the genomic regions that code for the upregulated pro-apoptotic genes?

6. Is there any evidence that would suggest that other chromatin remodeling enzymes (apart from BRG1 AND SMARCAL1) with Active DNA dependent ATPase domain would not be inhibited by ADAADi?

7. The details of statistical analysis done on the data sets in various experiments is incomplete or not available. The statistical test used to establish the significance and calculation of p-value should be written. The number of biological replicates which were used to generate the data should be mentioned in the figure legend. It is also surprising to see that less than 2-fold change in gene expression data has significant p-value.

8. Untrimmed images of the western blots or gels with appropriate standard marker should be made available in the supplementary section. This applies to all figures with western blot/gels.

Other Major Comments

Fig. 2

a) Is the quantification presented in Panel 2C is for the data presented in 2B? The data in 2B should be repeated to get an image where the bands are discrete and blot is not over saturated.

b) 2D & 2E- Was IP efficiency similar across the samples represented in these figures? Please provide appropriate data to support your response. Does ADAADi works as irreversible or competitive inhibitor? Is there a way to remove the bound ADAADi from the IPed enzyme before measuring its ATPase activity in vitro?

Fig. 6

a) Panels 6A to 6C- ADAADi leads to some decrease in the transcript levels of MMP-2 (less than 2 fold change). How does this small decrease in MMP-2 transcript levels leads to drastically reduced presence/activity of MMP-2 in the zymography assay (6B & C)? What would happen to MMP-2 activity (in zymography assay) for cells with ShBRG1 and ShSMARCAL1?

b) Panels 6D to 6E- Upon ADAADi treatment the percentage of apoptotic cells is approximately 15-20% (Sup Fig 3b and 3d). However, the efficiency of migration for ADAADi treated cells (as determined by wound healing assay) is 3 to 4 fold less as compared to untreated cells. How this discrepancy can be explained?

Minor comments

Supplementary Fig. 1

a) What was the rationale behind the selection of these 9 different cell lines?

b) The equation used for calculating the IC50 value should be provided in the methods section

c) Authors state that the IC50 at 48 hours is higher than IC50 at 24 hours. Is this statement correct?

d) In table 1, what is the criteria for designating certain cell lines as extremely sensitive or resistant? The equations used to reach to these conclusions should be described in the methods section.

Fig. 4

a) 4D & 4E- the quantified data in 4E is for 24 and 48 hr ADAADi treatment but the image in 4D is only for 48 hr ADAADi treatment. Authors should provide all the relevant data to make the figure complete.

Fig. 5 Maintain consistency in labelling of y-axis across all the panels

Fig7.

a) How was the colony area measured? Details should be provided in the methods section.

b) Hoechst staining will not overlay with the image presented for 12th day? Why this discrepancy is there?

6. PLOS authors have the option to publish the peer review history of their article (what does this mean?). If published, this will include your full peer review and any attached files.

Reviewer #1: No

Reviewer #2: No

Reviewer #3: No

---

## [Author Response · Author response to Decision Letter 0]

9 Feb 2021

Response to Reviewers Comments

We would like to thank the reviewers for their valuable suggestions. We have modified the manuscript as requested. We are providing point-by-point response to their comments,

Reviewer #1: In this paper, the authors explore the cellular effects of ADAADi, a chromatin remodeler ATPase inhibitor that is a metabolite of aminoglycosides.

I have several suggestions for improvement, but the overarching concern with the manuscript is a general lack of confidence in whether any of the phenotypes/gene changes observed are related to the inhibition of SMARCAL1 and BRG1. The evidence that these compounds inhibit ATPase function of SMARCAL1 and BRG1 is strong, but the fact that the compounds inhibit the growth of cell lines that don’t express SMARCAL1 and BRG1 calls the relevance of this function into question. If the conclusion of the manuscript is that ADAADi is a useful chemotherapy with unknown targets, that would be fine, as long as the data supports that conclusion. The pieces of data that do support that conclusion is the fact that ADAADi treatment doesn’t downregulate SMARCAL1 or BRG1 expression, and the general lack of correlation between gene expression changes (and miRNA changes) with ADAADi and shBRG1 or shSMARCAL1. The only data supporting ADAADi targeting BRG1/SMARCAL1 is the viability data in combination with knockdown; however, even that is problematic as any two agents that decrease cell growth would give similar results. Nevertheless, the discussion section still centers around ADAAdi acting as a BRG1/SMARCAL1 inhibitor, which is not supported in any way from the results. If the manuscript aims to define ADAADi as a potentially useful chemotherapy with unknown mechanism, there should be more results and discussion around this point.

Our Response: ADAADi binds to motif Ia present in ATP-dependent chromatin remodeling proteins [1]. As motif Ia is present in all ATP-dependent chromatin remodeling proteins, there is no a priori reason to believe that ADAADi will only target BRG1 and SMARCAL1. Indeed, we have previously shown that ADAADi inhibits the ATPase activity of Mot1 from Saccharomyces cerevisae, in addition to BRG1 and SMARCAL1 [2]. Therefore, ADAADi potentially can inhibit all ATP-dependent chromatin remodeling proteins. 

Cells that lack BRG1 and SMARCAL1 have other ATP-dependent chromatin remodeling proteins and treatment with ADAADi will inhibit them leading to cell death. 

We have now clarified this point in the revised manuscript.

Some possible avenues to pursue to create a more cohesive story:

Establish more definitively that ADAADi is not acting as a remodeling inhibitor. This could be through functional analysis (ATAC-Seq, ChIP-Seq, RNA-Seq). For example, I see the authors have performed ChIP-Seq for these factors (GSE137250), as have other labs. How does the RNA-Seq using ADAADi compare with BRG1/SMARCAL1 binding sites? Does ADAADi affect BRG1 binding and chromatin accessibility at these sites? Another possibility is to compare ADAADi with the Novartis BRG/BRM inhibitor (or the BI degrader), which has been shown functionally to affect BRG1 activity genome-wide. I know they are providing the compound for collaboration with academic labs (although with lots of MTA and hoops to jump through)

Our Response: ADAADi is a remodeling inhibitor. We have emphasized this point in the revised manuscript. We have previously shown that the nucleosomal remodeling activity of BRG1 is inhibited in the presence of ADAADi [2]. 

We have analyzed the ChIP-seq data (GSE137250) and have included the analysis in the revised manuscript. We have provided a table (Supplementary Table 4) listing the genes analyzed in the study that are regulated by BRG1/SMARCAL1 in the revised manuscript.

We also thank the reviewer for the suggestion to compare ADAADi with Novartis BRG1/BRM inhibitor. We would like to do these experiments in the future. We also plan to do ChIP-seq analysis in the presence of ADAADi to understand how the accessibility alters. At present our University remains closed due to the ongoing Covid-19 pandemic. Laboratory access is limited making it impossible for us to do any additional experiments at this point.

If the conclusion is that it is most likely not acting as a remodeling inhibitor, use the RNA-Seq data to develop hypotheses around what it is doing in cancer cells. There are many databases available for correlation to other inhibitors/processes. Are they affecting translation? The RNA-Seq indicates they might be. What is the activity of these metabolites on the ribosome?

Our Response: We do not know whether ADAADi directly inhibits the activity of ribosome, and thus, whether ADAADi directly affects the translation. However, it is always possible that ribosome activity is affected as a consequence of inhibiting the ATP-dependent chromatin remodeling proteins. Recently, Fun30, an ATP-dependent chromatin remodeling protein from S. cerevisiae, has been shown to regulate alternate splicing [3]. The RNA-seq data does indicate that alternate splicing is affected on ADAADi treatment. Our analysis showed that about 900 genes were switching to alternate transcripts after ADAADi treatment. We have not included this data in the manuscript. If the reviewer wishes, we can include it in. 

If the conclusion in the end is that they are good chemotherapies, I would add additional non-transformed cell lines to support this possibility.

Minor concerns

There are many more inhibitors of the SWI/SNF chromatin remodeling complexes now. While many of them inhibit chromatin targeting, there are two that act directly on BRG1: the ATPase inhibitor from Novartis, and the BRG/BRM degrader from BI. These two, at the very least, should be cited.

Our Response: We have included information about other inhibitors of the SWI/SNF chromatin remodeling complexes in the revised manuscript.

A panel displaying the compound structures referred to in the text (ADAADiN and ADAADiK) would be helpful.

Our Response: The structure of ADAADiN and ADAADiK has not yet been elucidated. We have, however, provided the structures of kanamycin and neomycin below. We have also shown the reaction catalyzed by Aminoglycoside phosphotransferase (APH) enzymes leading to the formation of ADAADi.

Kanamycin

https://pubchem.ncbi.nlm.nih.gov/compound/6032#section=2D-Structure

Neomycin

https://pubchem.ncbi.nlm.nih.gov/compound/Neomycin#section=2D-Structure

ADAADi reaction:

p. 13: It is a little confusing to say the IC50 value at 48 hours is “greater”, when it is actually smaller. Maybe say the inhibition was greater?

Our Response: We thank the reviewer for pointing out this error. We have now corrected it in the revised manuscript. 

Fig 1 D and E, the X-axis should be in log scale.

Our Response: We changed the X- axis to log scale in Fig. 1 D and E.

Reviewer #2: Radhakkrishnan et al report on the genome-wide effects of Active DNA-dependent ATPase inhibitor (ADAADi) in cultured mammalian cells. This inhibitor, derived from neomycin and a related inhibitor, ADAADiK, derived from kanamycin bind to ATP-dependent chromatin remodeling proteins, thereby inducing a conformational change and inhibiting activity. ADAADi was found to inhibit growth of mammalian cell lines, including cancer cells and promote apoptosis. This effect was dependent on BRG1 and SMARCL1 because depletion of these proteins desensitized cells to the drug. Although DNA damage was not observed, the expression of DNA damage response proteins was altered. Gene expression profiling revealed effects on expression of apoptotic protein coding genes and microRNAs. ADAADi also inhibited MMP2 and migration of cancer cells as did depletion of BRG1 and SMARCAL1.

Critique: This is the first comprehensive investigation of an inhibitor to the activity of ATP dependent chromatin remodeling enzymes. Although there are concerns about lack of selectivity, the findings that ADAADi can suppress growth and impede cancer migration suggest it may have therapeutic value. The study is well conducted and the findings convincing.

Minor Concerns

1. Figure legend 2B states that Westerns were done on both HeLa and DU145 cells, however, only HeLa cells are shown. Also, the text only indicates HeLa cells.

Our Response: We apologize for the mistake. We have corrected it in the revised manuscript. 

2. Much of the study associates ADAADi with inhibition of BRG1 and SMARCAL1 activity, yet the abstract does not even mention BRG1 and SMARCAL1. Some mention of the distinct and overlapping effects of ADAADi and depletion of BRG1 and SMARCAL1 by shRNA should be mentioned in the abstract. A little more discussion of this in the Discussion section would also be appropriate.

Our Response: ADAADi has the potential to inhibit all ATP-dependent chromatin remodeling proteins. SMARCAL1 and BRG1, in this study, were used as examples of ATP-dependent chromatin remodeling proteins. We have explained this point in detail in the revised manuscript. 

Reviewer #3: In the present study authors present an overview of the effect of ADAADi on the transcription profile of HeLa cells. Based on the RNA seq analysis of ADAADi treated cells, authors identify upregulation of pro-apoptotic genes as a major feature. They conducted a battery of standard experiments to suggest that ADAADi treatment leads to apoptosis, decrease in cell migration and attenuated cell growth. Based on the observations from their study authors propose that ADAADi has chemotherapeutic potential.

Overall, the study would appeal to the broad audience in cancer research and in principle can be good contribution to the field. However, the manuscript has substantial major issues:

1. Manuscript lacks clarity and does not convey a cohesive story. It lacks features of good organization and there is absence of harmony between the details described in the methods section and the results presented in the manuscript. Authors should ensure that the manuscript is well written with logical flow.

Our Response: We have revised and reorganized the manuscript extensively as per the reviewer’s suggestions. We hope that the revised manuscript presents a more cohesive story. 

2. Is there any evidence that the use of ADAADi on the non-cancerous cells (eg. fibroblasts) would not illicit similar change in transcriptional profile as being reported here for HeLa cells? If not, then authors should test the effect of ADAADi on transcriptome of non-cancerous cells as well as perform side by side comparison of cancerous vs normal cells in various assays reported here.

Our Response: We believe that any cell that takes up ADAADi will exhibit changes in the transcriptome because the inhibitor targets the chromatin remodeling proteins. We have not performed these experiments in non-cancerous cells. We plan to perform these experiments in the future.

3. Most of the experiments have been done using HeLa cells and with only ADAADiN at fixed sub-lethal concentration. However, manuscripts has statements suggesting that ADAADi induced transcriptional rewiring would be a common feature of various cancer cell lines. There is no experimental evidence to propose conservation of such phenomena. It would be interesting to see the effect of ADAADi on transcriptional modulation, and cell growth and migration in a few additional cancer cell lines.

Our Response: We had done these experiments on DU145 cells and have now provided the data in the revised manuscript. We find that the transcriptomic changes do occur in DU145 cells, but they are different as compared to HeLa cells. Previously we had shown that transcriptomic changes also occur in MDA-MB-231 cells on ADAADi treatment [4]. Further, in DU145 cells, we observe DNA damage, but the DNA damage repair is diminshed. DU145 cells too undergo apoptosis. Cell migration as observed by wound healing assay is retarded. Finally, we show that the anchorage independent growth is also inhibited. Thus, the phenomena observed on ADAADi treatment appears to be conserved. 

Apoptosis appears to be the general mechanism of cell death caused by ADAADi treatment as we have seen PC3 cells also undergo apoptosis on ADAADi treatment [5].

4. In many assays authors are comparing changes in transcript levels for different genes in ADAADi treated cells with the transcript levels of the same genes in cells treated with shRNA against BRG1 & SMARCAL1. Authors conclude that the trends in the two sets were not similar. Is this surprising given that treatment with ADAADi lead to upregulation of both BRG1 and SMARCAL1 whereas shRNA treatment would lead to decrease in level of both the proteins?

Our Response: We do indeed find that the transcript as well as protein expression of SMARCAL1 and BRG1 is upregulated at 48 hours of ADAADi treatment. However, the overexpressed protein is inactive as shown by the ATPase assays (Figure 2D and E). Therefore, the transcriptomic changes seen on downregulation of SMARCAL1 and BRG1 should have been similar to the overexpressed but inactive proteins present in HeLa cells after ADAADi treatment. But this was not observed.

5. Does BRG1 and SMARCAL1 regulate the transcription of impacted genes directly or indirectly? Do these chromatin remodelers localize to the genomic regions that code for the upregulated pro-apoptotic genes?

Our Response: We have shown experimentally that BRG1 and SMARCAL1 directly regulate ATM and ATR. These two proteins bind upstream of the transcription start site and regulate the expression on doxorubicin-induced DNA damage [6]. ChIP-seq analysis confirmed that BRG1 and SMARCAL1 are indeed present on the genomic regions of ATM and ATR. In the revised manuscript we have provided information regarding the occupancy of BRG1 and SMARCAL1 on the genomic regions of the genes analyzed in this study.

We have also analyzed the ChIP-seq data for the occupancy of BRG1 and SMARCAL1 on the genomic regions of upregulated pro-apoptotic genes. BRG1 and SMARCAL1 are not present on CASP3, CASP9, BCL-2 and BAX. Both are present on the genomic regions of PARP and Bcl2. We have presented this data as Supplementary Table 4 in the revised manuscript. 

6. Is there any evidence that would suggest that other chromatin remodeling enzymes (apart from BRG1 AND SMARCAL1) with Active DNA dependent ATPase domain would not be inhibited by ADAADi?

Our Response: ADAADi binds to motif Ia, which is present in all ATP-dependent chromatin remodeling proteins [1]. Previously, we had shown that ADAADi inhibits BRG1 as well as Mot1 from S. cerevisiae [2]. Therefore, we hypothesize that ADAADi can inhibit all ATP-dependent chromatin remodeling proteins. This is one of the reasons that ADAADi would be effective against all cancer cells. However, the effectiveness would depend upon the interaction between ADAADi and the protein. We do not have any data to support that the binding affinity is same or different for all the ATP-dependent chromatin remodeling proteins.

7. The details of statistical analysis done on the data sets in various experiments is incomplete or not available. The statistical test used to establish the significance and calculation of p-value should be written. The number of biological replicates which were used to generate the data should be mentioned in the figure legend. It is also surprising to see that less than 2-fold change in gene expression data has significant p-value.

Our Response: Statistical analysis was done using Sigma-plot. All experiments were done in triplicates. Average and standard deviation was calculated, and Student’s t-test was applied using Sigma-plot. We had mentioned it in the figure legends. In the revised manuscript we have also mentioned it in the methods section. The p value was calculated only for those genes where at least 25% change was observed consistently between biological and technical replicates. This is the reason for significant p value.

8. Untrimmed images of the western blots or gels with appropriate standard marker should be made available in the supplementary section. This applies to all figures with western blot/gels.

Our Response: We have provided the untrimmed images of all western blots and gels.

Other Major Comments

Fig. 2

a) Is the quantification presented in Panel 2C is for the data presented in 2B? The data in 2B should be repeated to get an image where the bands are discrete and blot is not over saturated.

Our Response: Yes, the quantification presented in Panel 2C is for the data presented in 2B. We understand the concern of the reviewer. The main problem is that the SMARCAL1 expression is very low in cells and we have to load at least 150 �g of protein to visualize it via western blot. Under these conditions, our �-actin loading control is always overexposed. We have provided the original blot for this figure in Supplementary Fig. 7 and it can be observed that SMARCAL1 bands are very faint as compared to the loading control. 

We have done these experiments many times and observed that SMARCAL1 and BRG1 levels are upregulated at 48 hr as compared to 24 hr. However, the quantitation might not be accurate due to overexposed �-actin blots. If the reviewer wishes, we can remove Panel 2C from the manuscript.

b) 2D & 2E- Was IP efficiency similar across the samples represented in these figures? Please provide appropriate data to support your response. Does ADAADi works as irreversible or competitive inhibitor? Is there a way to remove the bound ADAADi from the IPed enzyme before measuring its ATPase activity in vitro?

Our Response: We use equal amount of protein for immunoprecipitation. After immunoprecipitation we cannot calculate the protein concentration because we use Protein G beads for pull down. Therefore, we use the entire immunoprecipitated protein for ATPase assays. Hence, we cannot calculate the efficiency in these experiments. However, we have done separate immunoprecipitations for studying protein-protein interactions. In that case, IP to Input ratio was approximately 2 for both BRG1 and SMARCAL1 IP as calculated from western blots.

The binding studies show that the protein can bind to ADAADi both in the absence and presence of DNA. Further, DNA can bind to ADAADi bound protein [1]. The kinetic data suggests that ADAADi is a competitive inhibitor [2]. However, we have never been successful in competing off ADAADi using excess DNA. Indeed, our initial experiments showed that once ADAADi is bound to the protein, it can never be removed. Therefore, we do not have a method to remove the bound ADAADi from the IPed enzyme. It would have been a nice confirmatory assay if we could have removed the bound ADAADi and shown that the activity is restored. 

Fig. 6

a) Panels 6A to 6C- ADAADi leads to some decrease in the transcript levels of MMP-2 (less than 2-fold change). How does this small decrease in MMP-2 transcript levels leads to drastically reduced presence/activity of MMP-2 in the zymography assay (6B & C)? What would happen to MMP-2 activity (in zymography assay) for cells with ShBRG1 and ShSMARCAL1?

Our Response: Protein expression is dependent on multiple factors: transcript level, transcript stability, translation, and protein stability. Therefore, there is no direct correlation between transcript level and protein expression/activity. There are instances when transcript level is reduced but protein expression is unchanged. Similarly, there are instances when transcript level is only marginally reduced but protein expression drastically altered. We have only investigated the transcript level. It is quite possible that processes other than transcription are also altered leading to drastically reduced expression and activity of MMP-2 in zymography assay.

We observe decreased transcript levels of MMP-2 in ShBRG1 and ShSMARCAL1 cells. We also observe invasion (by wound healing assay) is reduced in BRG1 and SMARCAL1 downregulated cells. Therefore, we hypothesize that the activity of MMP-2 by zymography assay would also be reduced in these cells as compared to the control. 

b) Panels 6D to 6E- Upon ADAADi treatment the percentage of apoptotic cells is approximately 15-20% (Sup Fig 3b and 3d). However, the efficiency of migration for ADAADi treated cells (as determined by wound healing assay) is 3 to 4-fold less as compared to untreated cells. How this discrepancy can be explained?

Our Response: Invasive property of cancer cells depends on many factors including matrix metalloproteinases. There are many proteins and non-coding RNA that affect invasion. We have not investigated the expression of all the factors involved in this process. While apoptosis can influence invasion, there is no direct correlation between apoptosis and invasion. 

Minor comments

Supplementary Fig. 1

a) What was the rationale behind the selection of these 9 different cell lines?

Our Response: All the selected cell lines were epithelial cells though from different tissue of origins. Thus, the comparison has been made only between cancer cells of epithelial type. We have provided this clarification in the revised document as Supplementary Table 3.

b) The equation used for calculating the IC50 value should be provided in the methods section

Our Response: We have provided the equation used by the software in the methods section of the revised manuscript.

c) Authors state that the IC50 at 48 hours is higher than IC50 at 24 hours. Is this statement correct?

Our Response: We have corrected this statement in the revised manuscript. The statement now reads: 

…the IC50 at 48 hours was lesser than at 24 hours, implying greater inhibition at the later time point…

d) In table 1, what is the criteria for designating certain cell lines as extremely sensitive or resistant? The equations used to reach to these conclusions should be described in the methods section.

Our Response: A cell line was considered as extremely sensitive or resistant if the IC50 could not be calculated by the software (https://www.aatbio.com/tools/ic50-calculator/). In case of extremely sensitive cells, we got more than 50% kill at the lowest concentration of ADAADi tested. In case of extremely resistant cells, 50% kill was never obtained at the highest concentration of ADAADi tested. For these cell lines, the IC50 was calculated by fitting the data to quadratic equation. We have, however, removed the terms extremely resistant and extremely sensitive from the revised manuscript.

Fig. 4

a) 4D & 4E- the quantified data in 4E is for 24 and 48 hr ADAADi treatment but the image in 4D is only for 48 hr ADAADi treatment. Authors should provide all the relevant data to make the figure complete.

Our Response: We apologize for the mistake. We have now removed the quantified data for the 24 hr treatment.

Fig. 5 Maintain consistency in labelling of y-axis across all the panels

Our Response: We thank the reviewer for pointing out the error. We have ensured consistency in labelling of y-axis across all the panels in Fig. 5. 

Fig7.

a) How was the colony area measured? Details should be provided in the methods section.

Our Response: A ROI was drawn around the colony and the area under the ROI was quantified using the software provided by Nikon. For each day, the area for 100 colonies was measured. We have provided this information in the methods section of the revised manuscript.

b) Hoechst staining will not overlay with the image presented for 12th day? Why this discrepancy is there?

Our Response: We have not overlaid the Hoechst staining with any of the images presented in Figure 7B. After taking final 12-day live-cell colony imaging, cell colonies were suspended in agar gel and fixed with methanol. We took only DNA specific Hoechst images of the colonies. The Hoechst stained panel is standalone and not merged with any images. We explained it in the methods section of the revised manuscript. 

References:

1. Dutta P, Tanti GK, Sharma S, Goswami SK, Komath SS, Mayo MW, et al. Global epigenetic changes induced by SWI2/SNF2 inhibitors characterize neomycin-resistant mammalian cells. PloS One. 2012;7: e49822. doi:10.1371/journal.pone.0049822

2. Muthuswami R, Mesner LD, Wang D, Hill DA, Imbalzano AN, Hockensmith JW. Phosphoaminoglycosides inhibit SWI2/SNF2 family DNA-dependent molecular motor domains. Biochemistry. 2000;39: 4358–4365. 

3. Niu Q, Wang W, Wei Z, Byeon B, Das AB, Chen B-S, et al. Role of the ATP-dependent chromatin remodeling enzyme Fun30/Smarcad1 in the regulation of mRNA splicing. Biochem Biophys Res Commun. 2020;526: 453–458. doi:10.1016/j.bbrc.2020.02.175

4. Wu Q, Sharma S, Cui H, LeBlanc SE, Zhang H, Muthuswami R, et al. Targeting the chromatin remodeling enzyme BRG1 increases the efficacy of chemotherapy drugs in breast cancer cells. Oncotarget. 2016;7: 27158–27175. doi:10.18632/oncotarget.8384

5. Muthuswami R, Bailey L, Rakesh R, Imbalzano AN, Nickerson JA, Hockensmith JW. BRG1 is a prognostic indicator and a potential therapeutic target for prostate cancer. J Cell Physiol. 2019;234: 15194–15205. doi:10.1002/jcp.28161

6. Sethy R, Rakesh R, Patne K, Arya V, Sharma T, Haokip DT, et al. Regulation of ATM and ATR by SMARCAL1 and BRG1. Biochim Biophys Acta Gene Regul Mech. 2018;1861: 1076–1092. doi:10.1016/j.bbagrm.2018.10.004

---

## [Decision Letter · Decision Letter 1]

26 Feb 2021

PONE-D-20-37077R1

ALTERING MAMMALIAN TRANSCRIPTION NETWORKING WITH ADAADi: AN INHIBITOR OF ATP-DEPENDENT CHROMATIN REMODELING

PLOS ONE

Dear Dr. Muthuswami,

Thank you for submitting your manuscript to PLOS ONE. After careful consideration, we feel that it has merit but does not fully meet PLOS ONE’s publication criteria as it currently stands. Therefore, we invite you to submit a revised version of the manuscript that addresses the points raised during the review process.

We look forward to receiving your revised manuscript.

Kind regards,

Srinivas Saladi, Ph.D.

Academic Editor

PLOS ONE

Journal Requirements:

Additional Editor Comments (if provided):

Can you please respond to comments of reviewer 1, (response letter).

Reviewers' comments:

Reviewer's Responses to Questions

**Comments to the Author**

1. If the authors have adequately addressed your comments raised in a previous round of review and you feel that this manuscript is now acceptable for publication, you may indicate that here to bypass the “Comments to the Author” section, enter your conflict of interest statement in the “Confidential to Editor” section, and submit your "Accept" recommendation.

Reviewer #1: (No Response)

Reviewer #2: All comments have been addressed

Reviewer #3: All comments have been addressed

2. Is the manuscript technically sound, and do the data support the conclusions?

Reviewer #1: No

Reviewer #2: Yes

Reviewer #3: Yes

3. Has the statistical analysis been performed appropriately and rigorously? 

Reviewer #1: No

Reviewer #2: Yes

Reviewer #3: (No Response)

4. Have the authors made all data underlying the findings in their manuscript fully available?

Reviewer #1: Yes

Reviewer #2: Yes

Reviewer #3: (No Response)

5. Is the manuscript presented in an intelligible fashion and written in standard English?

Reviewer #1: No

Reviewer #2: Yes

Reviewer #3: Yes

6. Review Comments to the Author

Reviewer #1: In this revised manuscript, the authors attempt to address the concerns of the reviewers by putting in additional pieces of data. Unfortunately, while these data address some of the minor concerns of the reviewers, they fail to address the bigger concerns regarding a cohesive story and overinterpretation of results. In fact, one may argue that the addition of what is primarily more inconclusive data has further cluttered the storyline, as the authors state that they are unable to perform most the experiments proposed by the reviewers. In addition, the response to the reviewers worries me that the authors don’t entirely understand the concerns in the first place.

This puts me in a difficult position. I believe strongly in publishing quality data, even negative data, without regard to impact or utility. I am strong believer in basic science, and I support the mission of journals like PlosOne to publish papers regardless of the topic or perceived impact. But in order to maintain the quality of the journal, we have to maintain certain standards for peer review. I understand from the response that additional experiments are out of the question and the authors want to publish their work as is, which has many good pieces of data in it. If that is the case, I think the authors need to sit down and completely rework this paper to only include data that is robust, and only make conclusions supported by the data. I spent some time trying to work out how to do that, and then realized that I can’t dedicate that kind of time and energy into rewriting manuscripts I am reviewing when I have my manuscripts from my own lab that I am supposed to be rewriting. I also can’t follow most of the manuscript towards the end and the minute discussion of individual genes up and down here and there so I’m not sure what to do with it.

But here is my first pass at a suggestion for reworking the manuscript:

1) Take anything about cancer out of the intro. You have no non-transformed lines and have no intention of including them. Therefore, you do not have any evidence that this is a chemotherapeutic option of any sort. You can put something in the discussion about the future goals of investigating specificity towards cancer considering all the anticancer-related phenotypes you observe, but you currently have no data. Instead the intro should read: ADAADis inhibit ATPase chromatin remodelers in vitro, and have activity in cells. Therefore, this study aims to further understand the activity of ADAADi in cells.

2) The compelling data is 1) growth activity against cell lines, and 2) the RNA-seq and follow up phenotypic validation. Make that the focus of the paper.

3) Move all the BRG1, SMARCAL1 data to the end. All the concerns I expressed last time regarding a lack of evidence that these compounds target BRG1 or SMARCAL1 in cell lines (or the fact that there appears to be strong evidence that the compounds were definitely not targeting BRG1 or SMARCAL1 in these cell lines) were dismissed by the authors based on previously published in vitro data, and nothing compelling (published or not published) was added to indicate to me that BRG1 or SMARCAL1 are the primary targets, or in fact targets at all. I still see no reason for the authors to continue to interpret their data in relation to BRG1 or SMARCAL1. This is the most problematic part of the paper. This is where the authors need to sit and have a good hard look at the data and concisely state the evidence for and against BRG1 and SMARCAL1 as targets in any of the cell types tested. I get that the authors really want BRG1 and SMARCAL1 to be the targets, but they have to be more objective about what the results actually indicate. It seems as though there is some overlap in gene targets with shBRG1/SMARCAL1 and ADAADi, and some overlap between BRG1/SMARCAL1 ChIP-Seq peaks and DEGs. Is any of that overlap significant? It seems pretty cherry-picked the way it is presented right now. Similarly, the authors never addressed my concerns regarding fig 1d and e, and whether BRG1 or SMARCAL1 knockdown affects viability alone. If the cells are barely alive to begin with, then of course adding drugs doesn’t make them worse. You can’t kill dead cells. It is unclear from the way the data is normalized. In fact, nothing seems to be “100%” so I’m not sure what the viability is even normalized to. Everything should be normalized to control cells with no inhibitor.

Reviewer #2: The revised manuscript is improved and well written. The figures are high quality. I do not have any more concerns.

Reviewer #3: (No Response)

7. PLOS authors have the option to publish the peer review history of their article (what does this mean?). If published, this will include your full peer review and any attached files.

Reviewer #1: No

Reviewer #2: No

Reviewer #3: No

---

## [Author Response · Author response to Decision Letter 1]

25 Mar 2021

Response to Reviewer’s comments:

This puts me in a difficult position. I believe strongly in publishing quality data, even negative data, without regard to impact or utility. I am strong believer in basic science, and I support the mission of journals like PlosOne to publish papers regardless of the topic or perceived impact. But in order to maintain the quality of the journal, we have to maintain certain standards for peer review. I understand from the response that additional experiments are out of the question and the authors want to publish their work as is, which has many good pieces of data in it. If that is the case, I think the authors need to sit down and completely rework this paper to only include data that is robust, and only make conclusions supported by the data. I spent some time trying to work out how to do that, and then realized that I can’t dedicate that kind of time and energy into rewriting manuscripts I am reviewing when I have my manuscripts from my own lab that I am supposed to be rewriting. I also can’t follow most of the manuscript towards the end and the minute discussion of individual genes up and down here and there so I’m not sure what to do with it.

But here is my first pass at a suggestion for reworking the manuscript:

1) Take anything about cancer out of the intro. You have no non-transformed lines and have no intention of including them. Therefore, you do not have any evidence that this is a chemotherapeutic option of any sort. You can put something in the discussion about the future goals of investigating specificity towards cancer considering all the anticancer-related phenotypes you observe, but you currently have no data. Instead the intro should read: ADAADis inhibit ATPase chromatin remodelers in vitro, and have activity in cells. Therefore, this study aims to further understand the activity of ADAADi in cells.

Our Response: We regret that the reviewer felt compelled to help us rework the manuscript but are greatly appreciative of the effort!! Many thanks for your perspective which we have utilized to modify the introduction as suggested.

2) The compelling data is 1) growth activity against cell lines, and 2) the RNA-seq and follow up phenotypic validation. Make that the focus of the paper.

Our Response: We have removed all references to SMARCAL1 and BRG1 except from the section “ADAADi treatment alters the expression of ATP-dependent chromatin remodeling proteins”. In this section, we have highlighted the changes in the transcripts encoding for ATP-dependent chromatin remodeling protein including SMARCAL1 and BRG1. We have also included the western blots as well as activity assays for SMARCAL1 and BRG1 in this section as we felt it was pertinent to explain why we did the RNA-seq experiments after 48 hr ADAADi treatment. 

Overall, we have attempted to put the focus on growth activity and RNA-seq data followed by phenotypic validation.

3) Move all the BRG1, SMARCAL1 data to the end. All the concerns I expressed last time regarding a lack of evidence that these compounds target BRG1 or SMARCAL1 in cell lines (or the fact that there appears to be strong evidence that the compounds were definitely not targeting BRG1 or SMARCAL1 in these cell lines) were dismissed by the authors based on previously published in vitro data, and nothing compelling (published or not published) was added to indicate to me that BRG1 or SMARCAL1 are the primary targets, or in fact targets at all. I still see no reason for the authors to continue to interpret their data in relation to BRG1 or SMARCAL1. This is the most problematic part of the paper. This is where the authors need to sit and have a good hard look at the data and concisely state the evidence for and against BRG1 and SMARCAL1 as targets in any of the cell types tested. I get that the authors really want BRG1 and SMARCAL1 to be the targets, but they have to be more objective about what the results actually indicate. It seems as though there is some overlap in gene targets with shBRG1/SMARCAL1 and ADAADi, and some overlap between BRG1/SMARCAL1 ChIP-Seq peaks and DEGs. Is any of that overlap significant? It seems pretty cherry-picked the way it is presented right now. Similarly, the authors never addressed my concerns regarding fig 1d and e, and whether BRG1 or SMARCAL1 knockdown affects viability alone. If the cells are barely alive to begin with, then of course adding drugs doesn’t make them worse. You can’t kill dead cells. It is unclear from the way the data is normalized. In fact, nothing seems to be “100%” so I’m not sure what the viability is even normalized to. Everything should be normalized to control cells with no inhibitor.

Our Response: We have moved the correlation between the expression of SMARCAL1/BRG1 and the response of the cell lines to the end. We have explicitly stated that there is no correlation between the levels of SMARCAL1/BRG1 and the response of a cell line to ADAADi. We have also stated that there is no correlation between the occupancy of SMARCAL1/BRG1 with alteration in gene expression on ADAADi treatment. Since we cannot improve the data with additional experiments, we have removed all the data pertaining to ShSMARCAL1 and ShBRG1 as the data presented was not showing correlation between the expression/activity of these proteins and alterations in gene expression. However, if the reviewer feels that we need to include these data in the paper, we are willing to do so.

---

## [Editor Report · Decision Letter 2]

26 Apr 2021

Altering mammalian transcription networking with ADAADi:  An inhibitor of ATP-dependent chromatin remodeling

PONE-D-20-37077R2

Dear Dr. Muthuswami,

We’re pleased to inform you that your manuscript has been judged scientifically suitable for publication and will be formally accepted for publication once it meets all outstanding technical requirements.

Kind regards,

Srinivas Saladi, Ph.D.

Academic Editor

PLOS ONE
---

## [Editor Report · Acceptance letter]

6 May 2021

PONE-D-20-37077R2 

Altering mammalian transcription networking with ADAADi:  An inhibitor of ATP-dependent chromatin remodeling 

Dear Dr. Muthuswami:

I'm pleased to inform you that your manuscript has been deemed suitable for publication in PLOS ONE. Congratulations! Your manuscript is now with our production department. 

Kind regards, 

on behalf of

Dr. Srinivas Saladi 

Academic Editor

PLOS ONE